# MLPs Learn In-Context on Regression and Classification Tasks

**William L. Tong & Cengiz Pehlevan**
School of Engineering and Applied Sciences
Center for Brain Sciences
Kempner Institute for the Study of Artificial and Natural Intelligence
Harvard University, Cambridge, MA 02138
`{wtong@g,cpehlevan@seas}.harvard.edu`

## Abstract

In-context learning (ICL), the remarkable ability to solve a task from only input exemplars, is often assumed to be a unique hallmark of Transformer models. By examining commonly employed synthetic ICL tasks, we demonstrate that multi-layer perceptrons (MLPs) can also learn in-context. Moreover, MLPs, and the closely related MLP-Mixer models, learn in-context comparably with Transformers under the same compute budget in this setting. We further show that MLPs outperform Transformers on a series of classical tasks from psychology designed to test relational reasoning, which are closely related to in-context classification. These results underscore a need for studying in-context learning beyond attention-based architectures, while also challenging prior arguments against MLPs' ability to solve relational tasks. Altogether, our results highlight the unexpected competence of MLPs in a synthetic setting, and support the growing interest in all-MLP alternatives to Transformer architectures. It remains unclear how MLPs perform against Transformers at scale on real-world tasks, and where a performance gap may originate. We encourage further exploration of these architectures in more complex settings to better understand the potential comparative advantage of attention-based schemes.

## 1 Introduction

The last few years have witnessed meteoric progress in neural language models. Catalyzed by the Transformer architecture and driven by a steady increase in scale, these aptly-named Large Language Models (LLMs) demonstrate unprecedented competence in drafting grammatical text, answering questions, summarizing content, generating creative output, and even reasoning through non-trivial puzzles (Bubeck et al., 2023; Brown et al., 2020; Achiam et al., 2023).

Crucial to an LLM's proficiency is its ability to learn in-context (Lu et al., 2023; Dong et al., 2022; Brown et al., 2020). In-context learning (ICL) refers to a task paradigm where exemplars from a novel task are presented during inference time rather than during training (Figure 1a). The model must then respond correctly to a query based only on these novel exemplars. No weight updates occur throughout this process; rather, the model infers the task from the input exemplars and, despite having fixed weights, produces the correct output.

ICL is commonly assumed to be a unique ability of Transformers, and explanations of the phenomenon often ground their constructions in attention-based architectures (Akyürek et al., 2024; Von Oswald et al., 2023; Zhang et al., 2023; Reddy, 2024; Lu et al., 2024). On controlled regression and classification tasks targeted specifically for evaluating ICL, we demonstrate that the simple multi-layer perceptron (MLP) can also learn in-context — and moreover, *learn in-context competitively with the full Transformer given the same compute budget.*[1] These results suggest that ICL is not an exclusive

---

[1] Universal approximation by MLPs suggests that they may be able to learn in-context, though at uncertain cost. We demonstrate that MLPs learn in-context without significantly larger compute (and occasionally quite a bit smaller) than Transformers.

feature of attention-based architectures, and highlights the need for studying the phenomenon in a broader setting.

**Tasks.** We focus on controlled tasks commonly studied in the ICL literature, where the specific capacity for in-context learning can be precisely characterized. These tasks are necessarily synthetic approximations of natural language ICL prompting, but allow us to disambiguate a model's capacity for in-context learning from its ability to attain natural language fluency. In Section 2, we examine ICL versions of regression (Garg et al., 2022; Akyürek et al., 2024; Raventós et al., 2024; Zhang et al., 2023) and classification (Reddy, 2024; Chan et al., 2022). As the two primary task paradigms of machine learning, regression and classification form a representative basis for measuring ICL competency. In Section 3, we consider a series of classical tasks used in the psychology literature to probe relational reasoning (Campbell et al., 2023; Skinner, 1950; Sablé-Meyer et al., 2021), which are functionally in-context classification. On these tasks, we find that MLPs *outperform* Transformers, challenging common beliefs about MLPs' proficiency at relational reasoning (see Appendix A for an extended discussion). In focusing on controlled tasks, we avoid confounds irrelevant to ICL introduced by naturalistic settings like language and vision. Nonetheless, our findings remain consistent with existing results that *do* test MLPs in these more complex domains (Tolstikhin et al., 2021; Liu et al., 2021; Fusco et al., 2022; Bachmann et al., 2024).

**Ground rules.** To ensure different architectures are comparable across tasks, we observe the following ground rules. First, we compare models based on the total compute required for training (measured in peta-floating point operations, PFLOPs), which summarizes influences like parameter count, training iterations, and architectural efficiency. Details on how we compute this quantity are provided in Appendix C.13. Measuring by compute reflects the practical use of these models, fairly compares architectures by performance per floating-point cost, and is an established scale for defining neural scaling laws (Kaplan et al., 2020). Second, where a single model is required, we select the best model configuration as measured by loss, keeping compute cost equal across architectures. Data are presented online, reflecting the "one-pass" setting common in training large language models (Brown et al., 2020). Specific model and task configurations are enumerated in Appendix C.

## 1.1 RELATED WORK

In-context learning has been widely studied in a number of controlled settings. In particular, ICL has been reproduced for linear regression, where a Transformer trained to perform the task can extrapolate to novel input/label pairs provided in-context (Garg et al., 2022; Akyürek et al., 2024; Raventós et al., 2024; Wu et al., 2024; Bai et al., 2024; Li et al., 2023; Lu et al., 2024). Proposed mechanisms whereby a Transformer accomplishes the feat include that the Transformer implements some form of gradient descent (Von Oswald et al., 2023; Akyürek et al., 2024) or recapitulates least-squares or Ridge regression (Zhang et al., 2023; Akyürek et al., 2024; Lu et al., 2024). It has also been observed that a Transformer interpolates between *in-weight learning* (IWL), the traditional paradigm where the model learns specific examples through training, to in-context learning, where the model uses only exemplars provided in the input context at inference time (Raventós et al., 2024; Wu et al., 2024). Such a transition occurs as a function of *data diversity*, where datasets with more distinct examples encourage the development of ICL competency. Analogous phenomena have been observed in in-context classification tasks (Chan et al., 2022; Reddy, 2024). Impressively, the ICL performance attained in these tasks by Transformers approaches Bayes optimality (Xie et al., 2021; Bai et al., 2024; Li et al., 2023; Ahuja et al., 2024; Lu et al., 2024).

These studies nearly all ground their investigations in Transformer models, and explicitly assume that the model uses an attention mechanism to implement ICL. The exceptions include Chan et al. (2022), who discover that recurrent neural networks (both vanilla RNNs and LSTMs) are unable to learn an in-context classification task under the same conditions where a Transformer can, and Xie et al. (2021), who discover that LSTMs *can* in fact learn in-context on a synthetic language modeling task. Recently, Lee et al. (2024) found that a wide variety of causal sequence models can learn in-context on a broad array of toy tasks, with varying degrees of success. Park et al. (2024) support this finding by showing how state space models and their hybrids with Transformers can learn in-context competitively. To the best of our knowledge, no prior work has examined in-context learning in vanilla MLPs.

The current resurgence of interest in applying MLPs to modern, complex tasks originates with Tolstikhin et al. (2021), which introduced the MLP-Mixer model. Mixers operate by alternating MLPs across the dimensions of the input, treating the remaining dimensions as batch dimensions. Despite their simplicity, Mixers attain state-of-the-art performance on image classification, recalling the broad observation that "less inductive bias is better" (Sutton, 2019; Bachmann et al., 2024). In the ensuing years, "all-MLP" models based primarily on MLP components have spread across many areas including vision (Bachmann et al., 2024) and natural language (Liu et al., 2021; Fusco et al., 2022). While strong performance has been documented on natural language, less is known about MLPs' specific proficiency for ICL, and how it compares with Transformer models. In this study, we select a series of controlled, representative tasks that clarify an MLP's surprising competence for ICL. Our findings underscore the ultimate utility of MLPs, uncovering avenues of both theoretic and practical interest.

## 2 EXPERIMENT: IN-CONTEXT TASKS

We begin by exploring MLPs' behavior in a controlled ICL format, where their specific capacities and weaknesses can be precisely characterized. Specifically, we examine two tasks: in-context regression and in-context classification.

### 2.1 ICL REGRESSION

We present in-context regression following its common formulation (Garg et al., 2022; Zhang et al., 2023). The input consists of a sequence of values $(\boldsymbol{x}_1, y_1), (\boldsymbol{x}_2, y_2), \ldots, (\boldsymbol{x}_L, y_L)$, where $\boldsymbol{x}_i \in \mathbb{R}^n$ and $y_i \in \mathbb{R}$. The $\boldsymbol{x}_i, y_i$ pairs are linearly related through a set of weights $\boldsymbol{\beta} \in \mathbb{R}^n$ such that $y_i = \boldsymbol{x}_i \cdot \boldsymbol{\beta} + \varepsilon$, with noise $\varepsilon \sim \mathcal{N}(0, \sigma^2)$. Finally, the input includes a query $\boldsymbol{x}_q$. The model output is a single scalar regressed against the corresponding $y_q$. Crucially, the weights $\boldsymbol{\beta}$ vary between input sequences. The model cannot rely on learning any one $\boldsymbol{\beta}$. Rather, it must infer from context exemplars $(\boldsymbol{x}_i, y_i)$ what the corresponding $\boldsymbol{\beta}$ must be, and use this to predict the correct output $y_q$. Figure 1b illustrates the task, with additional details in Appendix C.

In the main text, our task fixes the number of context points at $L$. A common variation on this tasks allows the number of context points to vary, and trains the model autoregressively. Results on this autoregressive variation are presented in Figure 5, and are unchanged from the fixed-length case.

Following Raventós et al. (2024), we consider two different task distributions: finite and unrestricted. For the *finite* distribution, we fix a finite pool of weights before training $\boldsymbol{\beta}_1, \boldsymbol{\beta}_2, \ldots, \boldsymbol{\beta}_k$, where $\boldsymbol{\beta}_i \sim \mathcal{N}(\boldsymbol{0}, \boldsymbol{I}/n)$. For each input, we sample a new $\boldsymbol{\beta}$ by selecting uniformly at random one weight from the pool $\{\boldsymbol{\beta_i}\}_{i=1}^k$. Larger $k$ corresponds to higher data diversity. For the *unrestricted* distribution, a new set of weights is sampled for each input $\boldsymbol{\beta} \sim \mathcal{N}(\boldsymbol{0}, \boldsymbol{I}/n)$. The unrestricted distribution can be thought of as the $k \to \infty$ case, and requires full ICL competency in order to infer the correct weights relating the context exemplars. Unless otherwise stated, we use $n = 8$ dimensional inputs.

**Results.** We first consider how MLPs perform compared to Transformers on in-context regression. To do so, we train and test using online samples drawn from the unrestricted task distribution, requiring all models to learn an in-context solution. Figure 1c plots the MSE achieved by different architectures as a function of total compute. With sufficient compute, MLPs, Mixers, and Transformers *all* perform in-context regression with near optimal MSE, which is given by Ridge regression on context points using the Bayes optimal regularization parameter (Appendix C.6). For smaller compute, Transformers attain somewhat better MSE than their MLP counterparts, though the difference is modest and performance across all three architectures overlaps substantially.

One domain in which a vanilla MLP is decisively worse than a Transformer is for long context length. Figure 1d plots the excess MSE obtained after training and testing on the unrestricted task distribution for varying number of points in the context, where {excess MSE} = {model MSE} - {Bayes optimal Ridge MSE}. The Transformer generally approaches the optimal MSE regardless of context length, though it performs with less stability for longer contexts. The vanilla MLP worsens quickly with larger contexts and approaches the performance of an estimator that returns zero for every input. Strikingly, the MLP mixer does not exhibit the same sensitivity to context length, and continues attaining the Bayes optimal MSE consistently even for very long contexts.

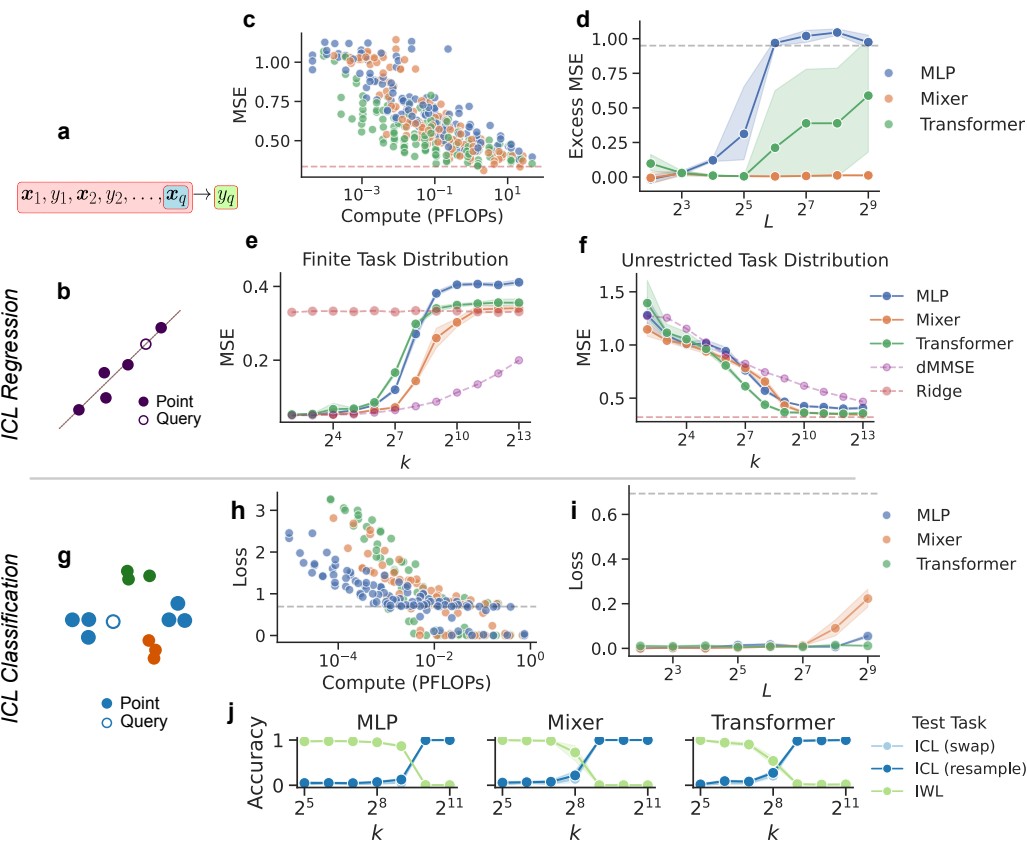

Figure 1: **ICL regression and classification results. (a)** ICL presents context exemplars from a novel task (*red*), followed by a query input (*blue*). The model must infer the solution (*green*) based on the context. **(b)** ICL regression example. The model receives linearly-related input points, and must regress the query point. **(c)** Compute vs. MSE on the unrestricted task distribution. Each point represents a single model, with particular parameters and training iterations. At large compute, MSE is approximately equal across all architectures. The red line corresponds to the Bayes optimal Ridge MSE. **(d)** Excess MSE (MSE above Bayes optimal) for varying context length $L$ on the unrestricted task distribution. Excess MSE remains flat for Mixers, but rises somewhat for Transformers. MLPs fail to learn in-context at all beyond $2^6$ context exemplars. The grey line corresponds to the excess MSE incurred by always guessing zero. **(e, f)** IWL to ICL transition with increasing data diversity. We train on a finite distribution with $k$ weights, then test on both the finite training distribution and the unrestricted distribution. All models exhibit a transition from IWL (represented by dMMSE) to ICL (represented by Ridge) as $k$ increases. Note: it is possible to "outperform" Bayes optimal Ridge on the finite training distribution by learning in-weight the underlying $\boldsymbol{\beta}$'s. **(g)** ICL classification example, with burstiness $B = 3$. Multiple clusters may share the same label. **(h)** Compute vs. cross entropy loss on ICL classification, with $k = 2048$ clusters, $B = 4$, and $L = 8$, which pushes all models to learn in-context. At large compute, all architectures attain near-zero cross entropy loss. The gray line corresponds to loss obtained from placing equal probability on the 2 (of $C = 32$) labels present in context. **(i)** Cross entropy loss for varying context length $L$ on the task configuration in (h). Loss is relatively flat for all architectures, though it increases a little for Mixers. **(j)** IWL to ICL transition with increasing data diversity, where $L = 8$ and $B = 4$. All models exhibit a transition from IWL to ICL as the number of clusters $k$ increases. **(all)** We use $n = 8$ dimension inputs. All line plots feature 95 percent confidence intervals about the mean, estimated from 5 replications.

One final observation: as data diversity increases, Transformers exhibit a transition from *in-weight learning* (IWL), the traditional paradigm where the model learns specific examples through training, to *in-context learning*, where the model uses only context exemplars presented at inference time (Raventós et al., 2024). We next show that MLPs exhibit a similar transition. Following Raventós et al. (2024), we train each model on a finite distribution with $k$ fixed regression weights. As we increase $k$, we record the MSE obtained by each model on both the finite distribution $\boldsymbol{\beta} \sim \mathcal{U}\left(\{\boldsymbol{\beta_i}\}_{i=1}^{k}\right)$ using the same $\boldsymbol{\beta}$'s from training (Figure 1e) and the unrestricted distribution $\boldsymbol{\beta} \sim \mathcal{N}(\mathbf{0}, \boldsymbol{I}/n)$ where $\boldsymbol{\beta}$'s are sampled anew (Figure 1f). We determine whether a model has learned the in-weight solution by comparing its MSE to that of the discrete minimum mean squared error (dMMSE) estimator, which is a Bayesian estimator derived from a prior matched to the finite training distribution (see Appendix C.6 for details).[2] We characterize the in-context solution by a Ridge estimator with the Bayes optimal choice of regularization. For small $k$, all models demonstrate in-weight learning by tracing the dMMSE curve. As $k$ increases, we observe a swift transition to the Ridge curve, indicating a transition to in-context learning. The Transformer makes this transition at a somewhat smaller $k$ than the MLP models. We consider additional plots and parameterizations in Appendix D.

## 2.2 ICL CLASSIFICATION

Following Reddy (2024), we present in-context classification as follows. The input consists of a sequence of context exemplars $(\boldsymbol{x}_1, \boldsymbol{y}_1), (\boldsymbol{x}_2, \boldsymbol{y}_2), \ldots, (\boldsymbol{x}_L, \boldsymbol{y}_L)$ followed by a query point $\boldsymbol{x}_q$, where $\boldsymbol{x}_i, \boldsymbol{y}_i \in \mathbb{R}^n$. The $\boldsymbol{x}$ points are sampled from a Gaussian mixture model $\mathcal{M}_k$ consisting of $k$ components. Each mixture component (i.e. cluster) is labeled by one among $C$ labels, where $k \geq C$, so multiple clusters may map to the same label. Labels are represented in the context by vectors $\boldsymbol{\alpha}_1, \boldsymbol{\alpha}_2, \ldots \boldsymbol{\alpha}_C \in \mathbb{R}^n$. If $\boldsymbol{x}_i$ belongs to cluster $j$, then $\boldsymbol{y}_i = \boldsymbol{\alpha}_j$. The model must predict the correct label for $\boldsymbol{x}_q$, and outputs $C$ logits corresponding to the $C$ labels (**not** a vector of values $\boldsymbol{\alpha}$, which are used only to represent labels in the context). Figure 1g illustrates this task, with additional details in Appendix C.7.

Importantly, the query point $\boldsymbol{x}_q$ shares a cluster with at least one of the context points $\boldsymbol{x}_1, \boldsymbol{x}_2, \ldots, \boldsymbol{x}_L$. Mixture components and cluster labels remain fixed throughout training. Hence, the model can learn either an in-weight solution by memorizing the label for each cluster, or an in-context solution by referencing the class label associated with $\boldsymbol{x}_q$ among the context exemplars. We also consider the input's *burstiness* $B$, which is the number of repeats per cluster in the context ($B$ must divide the context length $L$). For example, $B = 3$ means there are exactly three points from each cluster represented in the inputs. Data diversity corresponds to the number of clusters $k$, where larger $k$ correspond to more diverse dataset. Unless otherwise stated, we use $n = 8$ dimensional inputs.

**Results.** We first compare how different architectures perform at ICL across different compute in Figure 1h. The task is parameterized by burstiness $B = 4$ and $k = 2048$ with $L = 8$ points in the context, a setting in which all models develop an in-context solution (see Figure 7d for details). As before, with sufficient compute Transformers do not outperform vanilla MLPs or Mixers. Indeed, at small compute, vanilla MLPs attain a somewhat lower loss. Note: in this setting, there are $L/B = 2$ labels present in each context, out of $C = 32$ total possible labels. As a baseline, we plot in gray the loss obtained by placing equal probability on the 2 labels present in-context. MLPs in particular appear to plateau at this baseline before approaching zero loss with higher compute.

We also measure how well each model handles increasingly long context lengths in Figure 1i. In a surprising reversal, vanilla MLPs attain a relatively flat loss across context lengths, as do Transformers. Mixers' loss increases somewhat for longer contexts, though this blip vanishes at higher dimensions (Figure 7b). Overall, MLPs continue to perform at a comparable level with Transformers on in-context classification.

Finally, we observe a transition from IWL to ICL across the three architectures as data diversity increases. As in Reddy (2024), we measure IWL by constructing test examples where the query point *does not* appear in the context. The only way the model correctly classifies these points is if it memorizes the mapping from cluster to label. To measure ICL, we consider two different strategies: 1) sample points from an entirely different mixture $\mathcal{M}'_k$, producing novel clusters, or 2)

---

[2]The dMMSE estimator averages across the $k$ weights in the finite training distribution based on their fit to the current context exemplars.

swap cluster/label mappings so that clusters are labeled differently than they were during training. Test examples from either strategy can only be solved if the model identifies cluster labels in-context, since the underlying cluster label assignment is different from training.[3] We plot accuracy across all three types of test examples in Figure 1j for increasing $k$, and observe a transition from IWL to ICL across all three model architectures. The transition happens for somewhat lower data diversity in Transformers and Mixers, and somewhat higher in vanilla MLPs. Additional plots and parameterizations are explored in Appendix D.

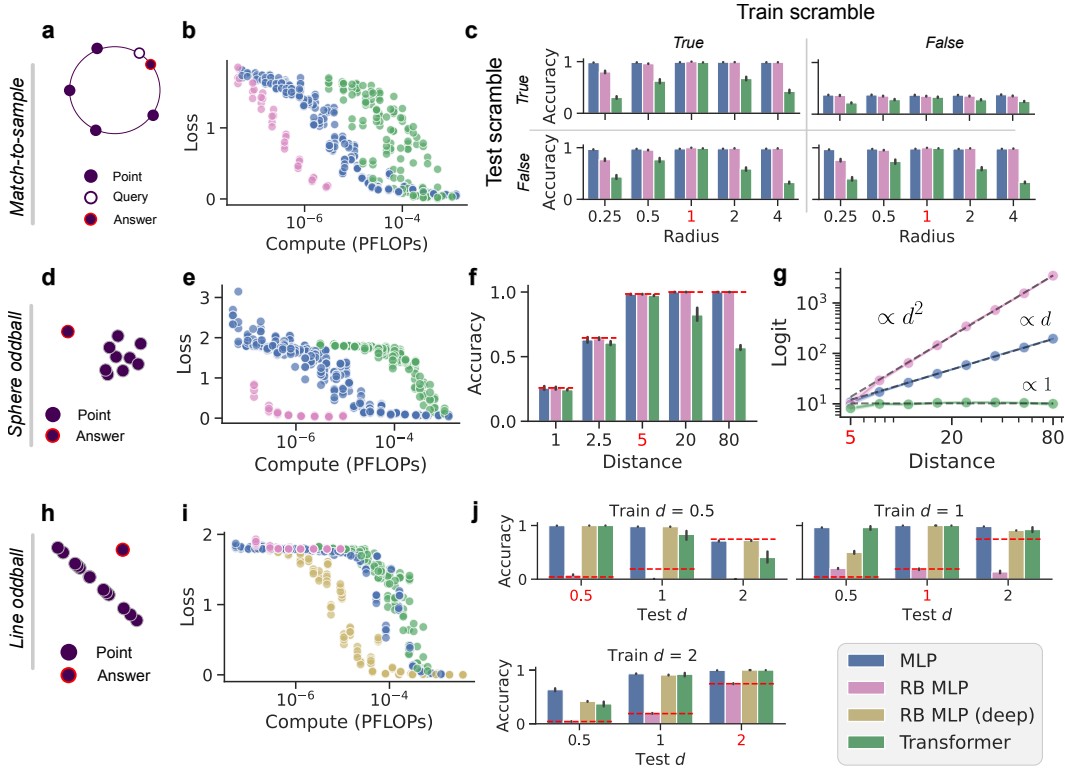

Figure 2: **Relational reasoning results.** Global legend is at the bottom right. **(a)** Match-to-sample task. **(b)** Compute vs. cross entropy loss on MTS task. Each point represents a single model, with particular parameters and training time. RB MLPs attain the best loss with the smallest compute, followed by MLPs and Transformers. **(c)** OOD generalization on MTS. In-distribution radii are highlighted in red. MLPs and RB MLPs generalize well on OOD radii. No model generalizes well on OOD test scrambling. **(d)** Sphere oddball task. **(e)** Same as in (b), for sphere oddball. **(f)** OOD generalization on sphere oddball. In-distribution distance is highlighted in red. Red dashed lines correspond to the accuracy obtained by guessing that the furthest point away from the cluster center is the oddball. **(g)** Logit of oddball point as its distance from the center increases. Dashed lines correspond to different polynomial scalings. Only the Transformer fails to increase its logit with distance. **(h)** Line oddball task. **(i)** Compute vs. loss on line oddball task. RB MLP no longer learns the task well, but appending additional MLP layers ("RB MLP (deep)") helps. **(j)** OOD generalization on line oddball. In-distribution distance is highlighted in red. Red lines indicate accuracy attained by a model guessing that the furthest point away from the center is the oddball. MLPs continue to generalize stronger than Transformers, and match the deep RB MLP. **(all)** Shaded regions and error bars correspond to 95 percent confidence intervals estimated from 5 replications.

---

[3]In practice, accuracy on these two ICL measures is virtually identical across all models and settings.

## 3 EXPERIMENT: RELATIONAL TASKS

We next consider classical tasks from psychology used to study relational reasoning in humans, animals, and neural networks (Campbell et al., 2023; Skinner, 1950; Sablé-Meyer et al., 2021; Geiger et al., 2023). These tasks are functionally a subset of in-context classification, and rely on understanding similarity relationships between context exemplars.

In a surprising advance from the tasks explored in Section 2, we find that MLPs perform *better* with *less compute* than Transformers, and generalize more effectively on out-of-distribution test sets. As a benchmark for gold-standard performance using hand-crafted features, we implement a relationally bottlenecked MLP (RB MLP), an architecture demonstrated to solve many challenging relational tasks with competitive generalization and efficiency characteristics (Webb et al., 2023; Campbell et al., 2023). Relational bottlenecks are architectural components that prevent absolute information about the inputs from propagating downstream; rather, the RB computes a set of (hand-crafted) relations between inputs (often simple dot products), and allows only these relations to flow downstream, forcing the model to operate on abstractions. Our RB MLP operates by first computing dot product relations between inputs, then propagating optionally through several MLP layers before a final readout (see Appendix C.5 for details). We find that while relational bottlenecks are helpful when the model's relations align well with the task structure, they may fail on tasks with deviating structure. All in all, these relational tasks demonstrate that MLPs can quite surprisingly outperform Transformers on certain in-context tasks.

The question of whether neural network models can reason relationally at all has been an enduring topic of heated debate (see Alhama and Zuidema (2019) for a recent review). Our results fall decisively in favor of the affirmative, and contrast a recent attempt at formally proving that MLPs cannot reason relationally (Boix-Adsera et al., 2023). In Appendix A, we discuss our relationship with the relational reasoning literature, comment on the proof by Boix-Adsera et al. (2023), and demonstrate empirically that a vanilla MLP solves a task posited by their arguments to be impossible.

### 3.1 MATCH TO SAMPLE

The match-to-sample (MTS) task paradigm originates in Skinnerian behavioral experiments (Skinner, 1950). A test subject is first presented with a sample stimulus (e.g. an image). The subject is then shown a set of many stimuli, and must select the one that matches the original sample.

Our MTS task proceeds as follows. The model is presented with $L$ context points $\boldsymbol{x_1}, \boldsymbol{x_2}, \ldots, \boldsymbol{x_L} \in \mathbb{R}^2$ followed by a query point $\boldsymbol{x}_q$. The context points are evenly spaced along a circle with unit radius centered at the origin. The model must return the index of the context point closest to the query $y = \arg\max_i (\boldsymbol{x}_i \cdot \boldsymbol{x}_q)$. The points can be thought of as idealized stimulus embeddings in neural representation space. A model must reason correctly about distances between points, and ignore their absolute location (which varies from example to example). Framed this way, the MTS setup is an in-context classification task with one context point per class. In the results that follow, we use $L = 5$ points in the context. Figure 2a presents an illustration of the task, with additional details in Appendix C.8.

In addition to the vanilla MLP and Transformer models, we also consider a relationally bottlenecked MLP architecture (RB MLP) (Webb et al., 2023). The RB MLP uses dot-product relations $\boldsymbol{r}$ between the query point and each of the five context points $\boldsymbol{r} = (\boldsymbol{x}_q \cdot \boldsymbol{x_1}, \boldsymbol{x}_q \cdot \boldsymbol{x_2}, \ldots, \boldsymbol{x}_q \cdot \boldsymbol{x_L})$. The relations $\boldsymbol{r}$ are passed directly to a softmax readout, producing class predictions $\boldsymbol{y}_{\mathrm{RB}} = \mathsf{smax}(\boldsymbol{W}_{\mathrm{readout}}\, \boldsymbol{r})$. Note, a choice of weights $\boldsymbol{W}_{\mathrm{readout}} = \boldsymbol{I}$ solves the task perfectly, though it remains to be seen whether the RB model discovers this solution. Further details on the RB MLP model are in Appendix C.5.

**Results.** Figure 2b plots the loss obtained by each of the three models on the MTS task as a function of compute. The vanilla MLP outperforms the Transformer by a surprising margin. With relations that are well-aligned to the task, the RB MLP model achieves the best compute efficiency.

We also consider how well each model performs in different kinds of out-of-distribution test examples. Results are plotted in Figure 2c. We first try perturbing the radius of the circle after training with unit radius. As we vary the radius during testing, both MLP models continue to perform well, but the Transformer quickly degenerates. We also try changing the order of context points. If the points are *ordered*, they are presented in clockwise order along the circle. If the points are *scrambled*, they

are presented in random order. Curiously, if the models are trained first on ordered points, then no model generalizes well when subsequently tested with scrambled points (not even the relationally bottlenecked model).

## 3.2 SPHERE ODDBALL

The oddball tasks described in the next two sections are based on work from Sablé-Meyer et al. (2021), who used it to measure geometric relational reasoning in humans and baboons. In an oddball task, the test subject is presented with six stimuli, originally geometric images. One image differs from the others. The subject must select this "oddball" to succeed. Like the MTS task, the oddball tasks are a subset of ICL classification where all-but-one point belong to the same class.

As before, our version of the oddball task simplify full visual stimuli into idealized stimulus representations. The model is presented with $L$ context points $\boldsymbol{x}_1, \boldsymbol{x}_2, \ldots, \boldsymbol{x}_L \in \mathbb{R}^2$. (There are no query points.) In the *sphere oddball* task, the context points are sampled as $\boldsymbol{x} \sim \mathcal{N}(\boldsymbol{\mu}, \boldsymbol{I})$, for some nonzero center $\boldsymbol{\mu}$. One point in the context is randomly selected and perturbed in a random direction by a distance $d$. The model must return the index of this oddball point. In the results that follow, we use $L = 6$ points in the context. Figure 2d presents an illustration of the task, with additional details in Appendix C.9.

In addition to the vanilla MLP and Transformer models, we again use an RB MLP with dot-product relations. Given the average context point $\overline{\boldsymbol{x}} = \frac{1}{L} \sum_i \boldsymbol{x}_i$, the relations $\boldsymbol{R}$ form a matrix with entries $R_{ij} = (\boldsymbol{x}_i - \overline{\boldsymbol{x}}) \cdot (\boldsymbol{x}_j - \overline{\boldsymbol{x}})$. These centered[4] dot-product relations are flattened and passed directly to a softmax readout, forming class predictions $\boldsymbol{y}_{\mathrm{RB}} = \mathsf{smax}(\boldsymbol{W}_{\mathrm{readout}} \, \mathsf{flat}(\boldsymbol{R}))$. Note, the sphere oddball task can be solved by finding the point that is furthest away from the cluster center. Hence, a choice of $\boldsymbol{W}_{\mathrm{readout}}$ that selects the diagonal relations $R_{ii}$ will solve the task, but it remains to be seen whether the model will learn such a readout. Additional details on the RB MLP are provided in Appendix C.5.

**Results.** Figure 2e plots the loss obtained by each model on the sphere oddball task as a function of compute. Again, the vanilla MLP outperforms the Transformer by a wide margin. With well-suited relations, the RB MP achieves the best compute efficiency.

We also consider how each model performs on OOD test examples. Training examples consist of oddballs with fixed perturbation distance $d = 5$. As we vary towards longer distances, we observe in Figure 2f that both the vanilla and RB MLPs continue performing perfectly, while the Transformer's performance decays. We can also examine how the logit corresponding to the oddball index changes as we change the position of the oddball with respect to the cluster center (Figure 2g). Both the vanilla and RB MLPs learn strictly increasing relationships, suggesting they will correctly generalize to any $d$ provided the other logits do not also increase. The Transformer seems to asymptote to a flat value, suggesting that it ultimately fails to distinguish the oddball logit for large $d$.

## 3.3 LINE ODDBALL

Rather than sample context points from a spherical Gaussian, the *line oddball* task distributes context points along a line. For each training example, we select a line with random orientation that passes through the origin. Context points $\boldsymbol{x}_1, \boldsymbol{x}_2, \ldots, \boldsymbol{x}_L \in \mathbb{R}^2$ are Gaussian distributed along this line with zero mean and unit variance. One context point is selected at random to be the oddball, and is perturbed by a distance $d$ in the direction perpendicular to the line. The model must output the index of the oddball point. We use $L = 6$ points in the context. Figure 2h presents an illustration of the task, with additional details in Appendix C. We use the same models as for the spherical oddball, including an RB MLP using the same relations $\boldsymbol{R}$.

The line oddball task cannot be solved by simply selecting the point furthest away from all the others for small $d$. The relevant relations are more sophisticated, and must be sensitive to the linear structure between context points. The line oddball task also presents an alternative hypothesis for the structure of stimuli in representation space. Whereas the sphere oddball presumes that input

---

[4]Centering was not required in the MTS task above, since the context points in that task were already centered.

stimuli are distributed isotropically in representation space, the line oddball task assumes that inputs fall along a favored direction. Neither is obviously more plausible than the other for a general task. However, as we see below, the RB MLP from the past two tasks fails quickly on this task, presenting a simple example in which well-aligned relations can be critical. We also experiment with a "deep" RB MLP, which features two additional MLP layers between the relations and readout, and now solves the task at a small compute cost.

**Results.** Figure 2i plots the loss for each model as a function of compute. Vanilla MLPs perform just a little better than Transformers. A (shallow) RB MLP fails to solve the task altogether, and loss remains high. A deep RB MLP, which features two additional fully-connected layers after the relational bottleneck, can solve the task.

We also compare how each model performs on different out-of-distribution test examples in Figure 2j. We vary the distance $d$ between the oddball point and the line of context points on different training sets. At test time, we compare the accuracy of each model on the whole range of $d$. As we saw above, MLPs continue to outperform Transformers on almost all OOD test cases. Unless equipped with further layers, the shallow RB MLPs fail to learn the task at all for small $d$. Though a relationally bottlenecked model can succeed with great efficiency on well-aligned tasks, without relations that are tailored to the task's underlying structure, an RB model may be disadvantaged.

## 4 DISCUSSION

We observed that MLPs can learn in-context and moreover, perform at a level comparable to Transformers on in-context tasks. We also examined relational reasoning tasks, closely related to ICL classification, which have historically been considered beyond the ability of simple neural architectures like MLPs (Alhama and Zuidema, 2019; Marcus, 1998; Boix-Adsera et al., 2023). Surprisingly, MLPs learn these relational tasks well — and exhibit both better compute efficiency and generalization performance than Transformers. This observation diverges from earlier claims (Boix-Adsera et al., 2023; Webb et al., 2023), but is consistent with the emerging realization that, given sufficient data diversity and compute, an MLP can indeed learn to reason relationally (Geiger et al., 2023). We discuss our relationship with prior work in relational reasoning further in Appendix A.

Broadly, our work is consistent with the "bitter lesson" of AI research (Sutton, 2019): in the face of increasing compute and data resources, general methods with weaker inductive bias will outperform specialist methods endowed with stronger inductive bias. This heuristic speaks to the intuition that a strong inductive bias may be beneficial for particular tasks, but may misalign the model in different or more general settings. We see an extreme example of this in studying relationally bottlenecked MLPs, where hand-crafted relations strongly benefit the model in specific cases where they align with the task. However, departing even slightly from the ideal task structure prevents the shallow RB MLP from learning the task at all, while a vanilla MLP continues to exhibit strong performance. In the absence of hand-designed relations, Transformers are more general learners than RB MLPs, but less than vanilla MLPs. As a result, for certain well-suited tasks (like ICL regression), Transformers perform more efficiently for a fixed compute budget. But for other tasks (relational reasoning, simple regression and classification in Appendix B), MLPs have the upper hand.

These results expand the range of possible tasks commonly thought solvable by MLP models. ICL may not be the exclusive domain of Transformers, and we encourage greater engagement with ICL beyond attention-based architectures. The surprising success of MLPs for relational reasoning also encourages a change in perspective about how simple architectures may be capable of solving relational tasks, and under what conditions they fail. The impressive performance of MLPs hints at potential real-world benefits, and we watch the future outcomes of all-MLP approaches with interest.

**Limitations and future directions.** We consider only controlled, synthetic tasks designed to probe for specific characteristics. We never compare architectures on complex datasets like those found in language or vision, though there are other studies that do, and find that MLPs continue to perform competitively (Tolstikhin et al., 2021; Liu et al., 2021; Fusco et al., 2022). The advantage of our approach is that we avoid confounds irrelevant to ICL introduced by complex data, and clarify the specific competencies of each model to learn in-context across representative settings. It remains overall unclear how MLP-based architectures perform against Transformers at scale in complex,

real-world tasks, and where a possible performance gap may originate. Future work should explore MLPs in more naturalistic settings to better understand the potential comparative advantage of attention-based schemes.

We also work exclusively in an online setting where models have access to a continuous stream of infinite data. As the bitter lesson heuristic predicts, this setup benefits the MLP, but we can certainly imagine that in data-limited scenarios, Transformers and other architectures with stronger inductive bias would dominate. Indeed, we have already observed that Transformers tend to learn in-context with comparatively less data diversity. Examining a data-limited setting represents another important future direction, and will potentially reveal important weaknesses in MLPs.

Where possible, we test on a wide array of parameterizations and task settings. The main text figures represent only an illustrative subset of our total results, with supplementary figures provided in Appendix D. However, as with any empirical study, we cannot test every infinite variation on our models and tasks; there may be further unexpected results hiding behind a setting we have not tried.

Overall, we hope our results encourage further work studying ICL beyond attention-based architectures, and the properties of simple architectures like MLPs that enable them to solve relational tasks. Important questions remain in quantifying how much data diversity is generally required to transition to ICL, how this threshold depends on architecture, varying sensitivity to context length across architectures, precisely characterizing differences in inductive bias for ICL, and more.

**Ethics statement.** Since this study focuses on synthetic tasks, it is limited in direct negative societal impacts beyond that of general theoretical machine learning research. We do not conduct experiments in human or animal subjects.

**Reproducibility statement.** Full descriptions of all tasks are provided in Sections 2 and 3 of the main text. Additional details regarding specific task and model configurations, along with code availability, compute requirements, and software, are comprehensively enumerated in Appendix C.

**Acknowledgements.** We thank Alex Atanasov, Hamza Chaudhry, Alex Meterez, Mo Osman, Sab Sainathan, Jacob Zavatone-Veth, members of the Pehlevan Group, and the anonymous ICLR reviewers for many helpful comments and discussions on our manuscript. WLT is supported by a Kempner Graduate Fellowship. CP is supported by NSF grant DMS-2134157, NSF CAREER Award IIS-2239780, DARPA grant DIAL-FP-038, a Sloan Research Fellowship, and The William F. Milton Fund from Harvard University. This work has been made possible in part by a gift from the Chan Zuckerberg Initiative Foundation to establish the Kempner Institute for the Study of Natural and Artificial Intelligence. The computations in this paper were run on the FASRC cluster supported by the FAS Division of Science Research Computing Group at Harvard University.

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

# A  MLPs REASON RELATIONALLY

Relational reasoning is closely related to in-context learning, as solving ICL tasks requires reasoning about the relationships between inputs while ignoring their absolute characteristics. Indeed, the relational tasks we explore in the main text are functional subsets of ICL classification. At the same time, MLPs are commonly presumed to *fail* at relational reasoning (Marcus, 1998; Boix-Adsera et al., 2023) or exhibit severe weaknesses (Fodor and Pylyshyn, 1988). While our primary focus remains on comparing in-context learning between Transformers and MLPs, we offer this digression to contextualize our results within the broader relational reasoning literature.

The question of whether connectionist models are able to reason relationally at all has been an enduring topic of passionate debate (see Alhama and Zuidema (2019) for a recent review). Our empirics support the notion that classical architectures like vanilla MLPs *can indeed reason relationally*, consistent with recent findings in Geiger et al. (2023). However, many researchers presuppose that classical architectures cannot solve relational tasks, resulting in a zoo of alternatives aimed at endowing neural networks with relational capacities (Webb et al., 2023; Battaglia et al., 2018; Geiger et al., 2023; Alhama and Zuidema, 2019).

One especially strong claim that MLPs cannot reason relationally was advanced by Boix-Adsera et al. (2023), who formally prove that MLPs will never generalize to unseen symbols on relational tasks. Their proof however relies on a pathological input scheme that hinders learning. Below, we discuss their analysis on MLPs and offer our own remarks on the learnability of relational tasks. We also demonstrate empirically that, under conventional settings, MLPs *do* generalize on a relational task claimed by Boix-Adsera et al. (2023) to be impossible.

## A.1  SUMMARY OF BOIX-ADSERA ET AL.

We begin with a brief summary of the relevant theorem in Boix-Adsera et al. (2023). Consider a *template $z$* consisting of a sequence of *wildcards*

$$z = \alpha_1 \alpha_2 \ldots \alpha_k \in \mathcal{W}^k \,.$$

A *string $x$* consisting of *symbols $x = x_1 x_2 \ldots x_k \in \mathcal{X}^k$* satisfies $z$ if there exists an injective map $s : \mathcal{W} \to \mathcal{X}$ such that $s(\alpha_i) = x_i$ for all $i$, which we call a *substitution map*. Finally, we have a labeling function $f^* : \mathcal{W}^k \to \mathbb{R}$ that assigns scalar labels to different templates.

A concrete example of this setup is the same-different task often used in probing relational reasoning (Geiger et al., 2023). Here, we consider two templates $z_1 = \alpha\alpha$ and $z_2 = \alpha\beta$. The labeling function is $f^*(z_1) = 1$ and $f^*(z_2) = 0$. The string $x = AA$ matches template $z_1$, and the string $x' = AB$ matches template $z_2$. We will abuse notation slightly and write $f^*(x) = f^*(z_1) = 1$. Crucially, matching a template depends only on relations between symbols and not on the symbols themselves. For example,

$$f^*(CC) = f^*(88) = f^*([\text{platypus}][\text{platypus}]) = 1 \neq f^*([\text{platypus}][\text{kangaroo}])$$

regardless of the concrete meaning of the symbols.

Boix-Adsera et al. (2023) consider MLP models with one-hot encodings of symbols

$$\boldsymbol{E}(\boldsymbol{x}) = (\boldsymbol{e}_{x_1}, \boldsymbol{e}_{x_2}, \ldots, \boldsymbol{e}_{x_k})^\mathsf{T}, \quad \boldsymbol{E}(\boldsymbol{x}) \in \mathbb{R}^{k \times |\mathcal{X}|}$$

where $\boldsymbol{e}_{x_i} \in \mathbb{R}^{|\mathcal{X}|}$ is a vector with 1 at an index corresponding to $x_i$ and 0 elsewhere. The one-hot encodings are then flattened as $\mathsf{flat}(\boldsymbol{E}(\boldsymbol{x})) \in \mathbb{R}^{k|\mathcal{X}|}$ before being passed directly into an MLP.

For notation, we write $f_{\text{MLP}}(\boldsymbol{x}; \boldsymbol{\theta}^t)$ as an MLP applied to string $\boldsymbol{x}$ with parameters $\boldsymbol{\theta}^t$ obtained after $t$ steps of stochastic gradient descent (SGD). We denote $\mathcal{X}_{uns}$ as symbols that are unseen during training, and $\mathcal{X}_{seen}$ as symbols that are seen during training. The theorem is stated as follows

**Theorem A.1** (From Boix-Adsera et al., failure of MLPs at generalizing on unseen symbols). *Suppose the label function $f^*$ is non-constant. Then for all SGD steps $t$, there exists a template $\boldsymbol{z} \in \mathcal{W}^k$ and a string $\boldsymbol{x}$ consisting of symbols $x_1 x_2 \ldots x_k \in \mathcal{X}_{uns}^k$ which satisfy $\boldsymbol{z}$ such that*

$$\mathbb{E}_{\boldsymbol{\theta}^t} \left[ \left( f_{MLP}(\boldsymbol{x}; \boldsymbol{\theta}^t) - f^*(\boldsymbol{z}) \right)^2 \right] \geq c > 0$$

*where $c$ is a constant that depends only on $f^*$, and the expectation is taken over random initialization of parameters $\boldsymbol{\theta}$ and subsequent SGD steps.*

Their proof relies on the permutation invariance property of MLPs and SGD (Ng, 2004). Summarizing briefly, they argue that if $x_1, x_2 \in \mathcal{X}_{uns}$, we can swap their one-hot encodings without any impact on the MLP. More generally, we can construct a permutation matrix $\Pi \in \mathbb{R}^{k|\mathcal{X}| \times k|\mathcal{X}|}$ such that for all strings of symbols $\boldsymbol{x}^{(1)}, \boldsymbol{x}^{(2)} \in \mathcal{X}_{uns}^k$ and $\boldsymbol{x}' \in \mathcal{X}_{seen}^k$, it remains true that $\Pi \mathsf{flat}(\boldsymbol{E}(\boldsymbol{x}^{(1)})) = \mathsf{flat}(\boldsymbol{E}(\boldsymbol{x}^{(2)}))$ and $\Pi \mathsf{flat}(\boldsymbol{E}(\boldsymbol{x}')) = \mathsf{flat}(\boldsymbol{E}(\boldsymbol{x}'))$. That is, we permute only the indices of unseen symbols, but leave the indices of seen symbols untouched. Then given the permutation symmetry of MLPs and SGD (Ng, 2004), because we preserve the indices of the seen symbols, we must have that

$$\mathbb{E}_{\boldsymbol{\theta}^t} f_{\mathrm{MLP}}\left(\boldsymbol{x}^{(1)}; \boldsymbol{\theta}^t\right) = \mathbb{E}_{\boldsymbol{\theta}^t} f_{\mathrm{MLP}}\left(\boldsymbol{x}^{(2)}; \boldsymbol{\theta}^t\right) .$$

Hence, if $f^*(\boldsymbol{x}^{(1)}) \neq f^*(\boldsymbol{x}^{(2)})$, the MLP cannot approach arbitrarily close to both labels, incurring an irreducible cost $c > 0$ that depends on the difference. In this way, MLPs cannot generalize to unseen symbols on any relational task.

## A.2 A DIFFERENT INPUT SCHEME

Is there a way to circumvent this impossibility result? One aspect of the proof that may seem suspect is its reliance on flattening one-hot encodings $\mathsf{flat}(\boldsymbol{E}(\boldsymbol{x}))$ as direct input to the MLP. Going as far back as Word2vec (Mikolov et al., 2013), a well-established convention for processing one-hot inputs is to instead pass them through an embedding matrix $\boldsymbol{W}_e$, creating vector embeddings

$$\boldsymbol{h}_0(\boldsymbol{x}) = (\boldsymbol{W}_e \boldsymbol{e}_{x_1}, \boldsymbol{W}_e \boldsymbol{e}_{x_2}, \ldots, \boldsymbol{W}_e \boldsymbol{e}_{x_k})^{\mathsf{T}}, \quad \boldsymbol{h}_0^{\pi}(\boldsymbol{x}) \in \mathbb{R}^{k \times d}$$

where $d$ is the dimension of a single symbol's vector embedding.[5] A practitioner then flattens and operates on the resulting vector embeddings, *not* the one-hot encodings directly. As we will shortly see, if we consider the more conventional input scheme that uses vector embeddings $\boldsymbol{h}_0(\boldsymbol{x})$ and *not* the one-hot encodings directly, then the conclusion from Boix-Adsera et al. (2023) no longer holds.

In particular, we consider an architecture where the input to the MLP is $\mathsf{flat}(\boldsymbol{h}_0(\boldsymbol{x}))$, rather than $\mathsf{flat}(\boldsymbol{E}(\boldsymbol{x}))$. We can attempt the same logic as before, and identify a permutation $\Pi$ such that for all strings of symbols $\boldsymbol{x}^{(1)}, \boldsymbol{x}^{(2)} \in \mathcal{X}_{uns}^k$ and $\boldsymbol{x}' \in \mathcal{X}_{seen}^k$, we have that $\Pi \mathsf{flat}(\boldsymbol{h}_0(\boldsymbol{x}^{(1)})) = \mathsf{flat}(\boldsymbol{h}_0(\boldsymbol{x}^{(2)}))$ and $\Pi \mathsf{flat}(\boldsymbol{h}_0(\boldsymbol{x}')) = \mathsf{flat}(\boldsymbol{h}_0(\boldsymbol{x}'))$. Unfortunately, if the embedding matrix $\boldsymbol{W}_e$ is randomly initialized like most neural network parameters, it is virtually impossible to find a permutation where $\Pi \mathsf{flat}(\boldsymbol{h}_0(\boldsymbol{x}^{(1)})) = \mathsf{flat}(\boldsymbol{h}_0(\boldsymbol{x}^{(2)}))$ while $\boldsymbol{x}^{(1)} \neq \boldsymbol{x}^{(2)}$. This is because the probability that any two elements of $\boldsymbol{W}_e$ are identical is zero for typical random matrix ensembles used in practice, e.g. if the elements of $\boldsymbol{W}_e$ are sampled i.i.d from a normal distribution.

Hence, it is clear that the original proof strategy of permuting the input, now $\mathsf{flat}(\boldsymbol{h}_0(\boldsymbol{x}))$, has become unviable. However, a skeptical reader might now wonder whether Theorem A.1 might still be saved if we apply permutations to the one-hot encodings *before* they are passed to the embedding matrix. That is, given a permutation matrix $\Pi \in \mathbb{R}^{|\mathcal{X}| \times |\mathcal{X}|}$, we construct

$$\boldsymbol{h}_0^{\pi}(\boldsymbol{x}) = (\boldsymbol{W}_e(\Pi \boldsymbol{e}_{x_1}), \boldsymbol{W}_e(\Pi \boldsymbol{e}_{x_2}), \ldots, \boldsymbol{W}_e(\Pi \boldsymbol{e}_{x_k}))^{\mathsf{T}} .$$

In this way, Theorem A.1 might still be rescued through permutations on one-hots before the embedding matrix. This method sidesteps the issue with permuting $\mathsf{flat}(\boldsymbol{h}_0(\boldsymbol{x}))$ directly, and the MLP trained on SGD remains invariant to any permutation on the underlying one-hots. Hence, it seems the proof may remain valid, and the impossibility result might still holds.

Unfortunately, this scheme runs into a different issue: it is impossible to find two inputs $\boldsymbol{x}^{(1)}, \boldsymbol{x}^{(2)}$ where $\Pi \boldsymbol{x}^{(1)} = \boldsymbol{x}^{(2)}$, but that $f^*(\boldsymbol{x}^{(1)}) \neq f^*(\boldsymbol{x}^{(2)})$. (Note, we have abused notation slightly and write $\Pi \boldsymbol{e}_x = \Pi \boldsymbol{x}$.) Indeed, we next show that if $\boldsymbol{x}$ satisfies a template $\boldsymbol{z}$, then any permutation $\Pi$ on the symbols of $\boldsymbol{x}$ will also satisfy $\boldsymbol{z}$. This can be seen quite simply by considering that 1) by definition, template satisfaction is invariant under a relabeling of symbols and 2) any permutation is a relabeling of symbols — hence, template satisfaction must be invariant under permutation. We phrase this formally below.

**Proposition A.1** (Permutation invariance of template satisfaction). *For any template $\boldsymbol{z} \in \mathcal{W}^k$ and any permutation $\Pi : \mathcal{X} \to \mathcal{X}$, if the string $\boldsymbol{x}$ satisfies $\boldsymbol{z}$, then $\Pi \boldsymbol{x}$ also satisfies $\boldsymbol{z}$.*

---

[5]Indeed, in their results on Transformers, Boix-Adsera et al. do use vector embeddings. It is unusual they would choose to omit them in their analysis of MLPs.

*Proof.* If symbols $\boldsymbol{x} = x_1 x_2 \ldots x_k$ satisfy the template $\boldsymbol{z} = \alpha_1 \alpha_2 \ldots \alpha_k$, then there exists an injective substitution map $s$ such that $s(\alpha_i) = x_i$. Because permutations $\Pi$ are bijective, there must also exist an injective substitution map $s'$ such that $s'(\alpha_i) = \Pi(x_i)$. Hence, $\Pi\boldsymbol{x}$ satisfies the template $\boldsymbol{z}$.

$\square$

In this way, it is not actually possible to find two strings $\boldsymbol{x}^{(1)}, \boldsymbol{x}^{(2)}$ such that $\Pi\boldsymbol{x}^{(1)} = \boldsymbol{x}^{(2)}$ but for which $f^*(\boldsymbol{x}^{(1)}) \neq f^*(\boldsymbol{x}^{(2)})$ since they both satisfy the same template. Permuting over one-hot encodings before the embedding matrix is not viable. Alternatively, we could try permuting the output from an intermediate layer of the MLP, but this will fail for the same reason that permuting $\mathsf{flat}(\boldsymbol{h}_0(\boldsymbol{x}))$ failed. All in all, if we replace the input $\mathsf{flat}(\boldsymbol{E}(\boldsymbol{x}))$ with the more conventional $\mathsf{flat}(\boldsymbol{h}_0(\boldsymbol{x}))$, Theorem A.1 is no longer valid.

### A.3 CAN MLPS REASON RELATIONALLY?

We have argued that the impossibility theorem of Boix-Adsera et al. (2023) can be circumvented, but it remains to be seen whether MLPs can truly reason relationally. We next identify coarse conditions that would in principle allow an MLP to generalize to unseen symbols given finite training data. Intuitively, we can imagine that if the MLP's training data is sufficiently diverse and the model is sufficiently expressive and smooth, then any unseen input $\boldsymbol{x}$ will fall somewhat close to a seen input $\boldsymbol{x}'$, so $f_{\mathrm{MLP}}(\boldsymbol{x}) \approx f_{\mathrm{MLP}}(\boldsymbol{x}') \approx f^*(\boldsymbol{x}')$. If $\boldsymbol{x}$ and $\boldsymbol{x}'$ are labeled the same (not unreasonable, if they are close), then the MLP would indeed be generalizing on an unseen input example.

We formalize this intuition in the following proposition, which establishes coarse conditions for a model $f$ to generalize on unseen input. We adopt the same setting as above, but we now treat strings $\boldsymbol{x}$ as real-valued $\boldsymbol{x} \in \mathbb{R}^n$. This is equivalent to flattening the vector embeddings $\boldsymbol{h}_0(\boldsymbol{x})$ generated from one-hot encoded symbols $x_1 x_2 \ldots x_k$. Doing so simplifies the following discussion.

**Proposition A.2** (Conditions for generalizing to unseen inputs). *Fix $\varepsilon$ and select $\delta < \varepsilon/3$. Given a model $f : \mathbb{R}^n \to \mathbb{R}$ and labeling function $f^* : \mathbb{R}^n \to \mathbb{R}$, if they satisfy the following three conditions*

1. ***Smoothness**: $f$ and $f^*$ are L-Lipschitz continuous*

2. ***Expressivity**: for all $\boldsymbol{x}$ that are seen, $|f(\boldsymbol{x}) - f^*(\boldsymbol{x})| < \delta$.*

3. ***Data diversity**: for all $\boldsymbol{x}'$ that are unseen, there exists an $\boldsymbol{x}$ that is seen such that $||\boldsymbol{x} - \boldsymbol{x}'|| < \delta/L$*

*then*

$$|f(\boldsymbol{x}) - f^*(\boldsymbol{x})| < \varepsilon$$

*for all $\boldsymbol{x}$ (seen and unseen).*

*Proof.* This statement is a simple consequence of the triangle inequality. For any unseen $\boldsymbol{x}'$ in the $\delta/L$-neighborhood of seen $\boldsymbol{x}$, we have that

$$|f(\boldsymbol{x}') - f^*(\boldsymbol{x}')| \leq |f(\boldsymbol{x}') - f(\boldsymbol{x})| + |f(\boldsymbol{x}) - f^*(\boldsymbol{x})| + |f^*(\boldsymbol{x}) - f^*(\boldsymbol{x}')|$$
$$\leq 3\delta \,.$$

Hence, if we select $\delta < \varepsilon/3$, we must have that $|f(\boldsymbol{x}') - f^*(\boldsymbol{x}')| < \varepsilon$. $\square$

In this way, if a model satisfies the above three conditions, it generalizes to unseen inputs for a task defined by the labeling function $f^*$. The first condition for smoothness is regularly achieved by standard neural networks (Khromov and Singh, 2023). The second condition corresponds to a notion of *expressivity* — that is, a model $f$ should be able to approach arbitrarily close to zero on its training data. For modern neural network models trained on simple tasks, this is a frequent occurrence (Zhang et al., 2021). The third condition corresponds to a coarse description of *data diversity*. The training data should be sufficiently diverse such that all unseen examples are very close to an example seen during training. This condition may be difficult to achieve in practice, but it offers a very coarse upper bound on the requisite data diversity required to generalize on unseen examples. Nonetheless, an MLP trained online on a suitably constrained input space may very well achieve this condition.

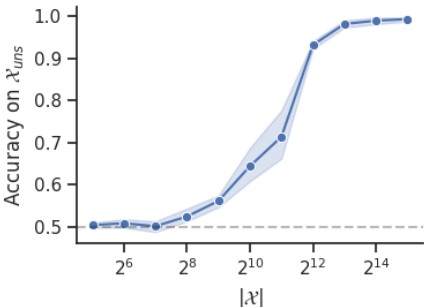

Figure 3: **MLP accuracy on unseen symbols for the same-different task.** The gray dashed line indicates chance-level performance. Shaded region indicates 95 percent confidence regions estimated from 5 replications. For higher data diversity (i.e. number of symbols in the task), the MLP generalizes progressively better. Beyond roughly $2^9$ symbols in the task, the MLP performs substantially above chance, and approaches perfect generalization beyond $2^{12}$ symbols.

Given further assumptions on $f$ (e.g. $f$ is an MLP), it is likely possible to shrink this data diversity bound considerably.

Regardless whether an MLP achieves these conditions exactly, we next show that with sufficient data diversity, an MLP equipped with an embedding matrix and trained through gradient descent *does* solve a relational task of the form posited in Theorem A.1, generalizing perfectly to unseen data.

## A.4 SAME-DIFFERENT TASK

We now demonstrate empirically that a vanilla MLP trained with gradient descent will discover a solution that generalizes to unseen symbols. While the tasks we explore in Section 3 already indicate that MLPs solve relational tasks, to address any remaining doubt, we pursue an ostensibly impossible task as specified in Boix-Adsera et al. (2023). In particular, we examine the same-different task.

As noted in Appendix A.1, the same-different task consists of two templates, $z_1 = \alpha\alpha$ and $z_2 = \alpha\beta$, with labels $f^*(\alpha\alpha) = 1$ and $f^*(\alpha\beta) = 0$. Following Boix-Adsera et al. (2023), we consider input strings $\boldsymbol{x} = x_1x_2 \in \mathcal{X}^2$ that are *one-hot encoded* before being passed through a randomly-initialized embedding matrix. As previously discussed, this embedding enables the model to circumvent the impossibility result (Theorem A.1) from Boix-Adsera et al. (2023). The remainder of our model is exactly the MLP described in Appendix C. We use an MLP with 4 hidden layers (in addition to an embedding layer) all of width 256 dimensions. The MLP is trained on batches of 128 examples sampled from a set of symbols with varying size $|\mathcal{X}|$, with roughly even positive and negative examples. In each run, the MLPs are first trained for $10,000$ batches on half the symbols in $\mathcal{X}$, then tested on the other half. All remaining hyperparameters are specified in Appendix C.

We plot the performance of an MLP on this task in Figure 3. For this task, data diversity refers to the number of symbols in $\mathcal{X}$. With higher data diversity, we see that the MLP improves progressively at generalizing on unseen symbols. Beyond about $2^{12}$ symbols, the MLP generalizes near-perfectly, confirming that these models do, indeed, learn to reason relationally. This result is particularly interesting because it shows that, with sufficient data diversity, the MLP *generalizes to completely novel symbols*.

## A.5 DISCUSSION

Our results are consistent with Geiger et al. (2023), who also find empirically that MLPs (among other architectures) reason relationally and generalize robustly to unseen inputs. We complement their results by further evidencing the possible conditions where MLPs may continue to generalize successfully. Geiger et al. (2023) argue that neural networks require "non-featural input representations" to generalize. A representation is featural if it encodes interpretable features of the task in axis-aligned dimensions. One-hot token encodings are featural, but randomized encodings are not.

As in Geiger et al. (2023), we show that featural representations like one-hot encodings remain usable provided they that pass through an embedding matrix, becoming non-featural and circumventing the impossibility result found by Boix-Adsera et al. (2023). In this way, with sufficient data diversity, an MLP still generalizes to unseen inputs, even if the inputs are unseen one-hot encodings.

Despite our success above, many earlier studies document cases where common neural network architectures fail to reason relationally (Marcus et al., 1999; Kim et al., 2018; Lake and Baroni, 2018; Alhama and Zuidema, 2019). One important reason for the failure may be that the task inputs are very large and complex, as in visual reasoning (Kim et al., 2018; Serre, 2019). Proposition A.2 suggests that the data diversity required for successful generalization scales exponentially with the dimension of the inputs in the worst case. It is possible that given a sufficiently vast dataset, an MLP *would* perform well on visual reasoning tasks. Furthermore, having shown above that MLPs are decisively capable of relational reasoning (especially when presented with idealized stimulus embeddings, as in Section 3), their failure on complex tasks highlights a need to separate a model's ability to reason relationally from its ability to learn sufficiently rich feature representations. In realistic data-limited scenarios, perhaps an MLP paired with a more bespoke module for feature learning would reason quite successfully. We anticipate further work that more closely investigates whether these failures stem from data limitations, insufficient feature learning, or some other cause, thereby building a more complete and updated picture of relational reasoning in neural networks.

## B EXPERIMENT: SIMPLE TASKS

In the main text, we showed that MLPs perform comparably with Transformers on ICL regression and classification, and better on relational tasks. In this separate set of experiments, we examine a setting in which MLPs are *decisively superior*. To do so, we depart from in-context tasks and consider simple (non-ICL) regression and classification.

### B.1 SIMPLE REGRESSION

Following the classic regression setup, the model receives as input a single point $x \in \mathbb{R}^n$, and must output the corresponding $y \in \mathbb{R}$ which is related through $y = x \cdot \beta$. **Note**: this is *not* in-context regression, so the model receives only a single input $x$ and the weights $\beta$ remain fixed throughout the duration of the task. For the Transformer, unless otherwise stated, each input coordinate is processed as a "token" with depth 1. Additional details are provided in Appendix C.11.

**Results.** In Figure 4a, we plot the MSE of vanilla MLPs and Transformers as a function of compute on $n = 64$ dimensional regression. The gap between the two models is substantial. The Transformer seems to struggle especially for larger inputs. For smaller $n$, the compute gap shrinks between MLPs and Transformers (Figure 10). If your are stuck with large $n$, one potential strategy for improving the Transformer's performance is to manually chunk the inputs into larger tokens, reducing the total number of tokens. In the extreme case, we chunk the entire input into a single token (effectively transposing the input). As the token size increases, the Transformer's effiency smoothly improves until it reaches a level comparable to the MLP (Figure 4b). Indeed, in the extreme case of a single input token, the Transformer is almost identical to an MLP anyway.

### B.2 SIMPLE CLASSIFICATION

We next consider a classic classification setup. The model receives a single point $x \in \mathbb{R}^n$ that was sampled from 1 of $k$ different clusters. The model must output the correct label $y$ of the corresponding cluster. This is *not* in-context classification, so the model receives only a single input $x$ and the cluster/label mapping remains fixed throughout the duration of the task. Additional details are provided in Appendix C.12.

**Results.** The same results continue to hold. As shown in Figures 4(c,d), for $n = 64$ dimensional classification, there is a wide compute gap between a vanilla MLP and a Transformer model, though the gap can be narrowed by manually chunking the inputs into larger tokens. Figure 10 gives performance for inputs of different dimensions, where smaller $n$ narrow the gap between the two models.

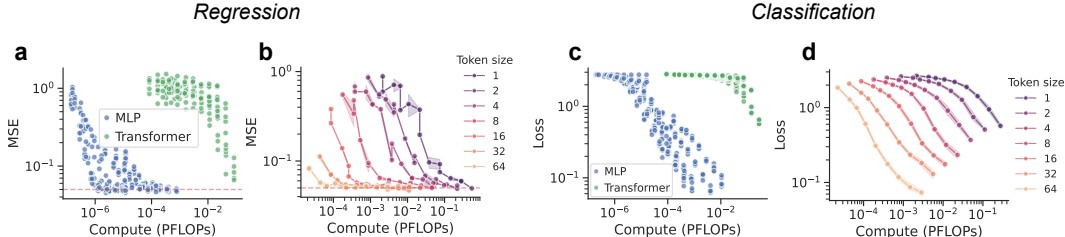

Figure 4: **Simple regression and classification results. (a)** MLPs attain substantially lower MSE at lower compute than Transformers. The red line corresponds to the minimum attainable MSE. **(b)** Transformers attain performance given larger token sizes. **(c, d)** Same as in (a, b), for classification, with $k = 16$ clusters. **(all)** We use $n = 64$ dimension inputs. Other parameterizations are explored in Appendix D. Shaded regions correspond to 95 percent confidence intervals estimated from 5 replications.

### B.3 DISCUSSION

Evidently simple tasks with long inputs work against the Transformer's attention mechanism. Shortening the context by reducing the task dimension, chunking inputs into larger tokens, or bypassing the attention mechanism altogether by stacking the input into a single token all improve the Transformer's efficiency. It is not immediately obvious why the Transformer performs so dramatically worse compared to the MLP for larger $n$, though it is well-known that Transformers can struggle with long inputs (Tay et al., 2020).

## C  MODEL AND TASK CONFIGURATIONS

In the following appendix, we provide all details on the specific model and task configurations used in this study, including architecture, hyperparameter settings, training methodology, and more.

### C.1  CODE

For the most precise information on our setup, please refer to our GitHub code repository:

https://github.com/wtong98/mlp-icl

There, you will find all code used to reproduce the plots in this document, as well as any minor implementation details omitted from this appendix. If you notice an error, we welcome your pull requests!

### C.2  MLP

The MLP accepts inputs $\boldsymbol{x} \in \mathbb{R}^n$. If a task provides inputs of shape $L \times D$ (length by token depth), the inputs are first flattened to size $n = LD$ before being passed to the MLP. A model with $\ell$ hidden layers then proceeds as follows:

$$\boldsymbol{h}_1(\boldsymbol{x}) = \phi\left(\boldsymbol{W}_1\boldsymbol{x} + \boldsymbol{b}_1\right)$$
$$\boldsymbol{h}_2(\boldsymbol{x}) = \phi\left(\boldsymbol{W}_2\boldsymbol{h}_1(\boldsymbol{x}) + \boldsymbol{b}_2\right)$$
$$\vdots$$
$$\boldsymbol{h}_\ell(\boldsymbol{x}) = \phi\left(\boldsymbol{W}_\ell\boldsymbol{h}_{\ell-1}(\boldsymbol{x}) + \boldsymbol{b}_\ell\right)$$
$$\boldsymbol{f}_{\text{MLP}}(\boldsymbol{x}) = \boldsymbol{W}_{\text{out}}\boldsymbol{h}_\ell(\boldsymbol{x}) + \boldsymbol{b}_{\text{out}}$$

For all tasks, we use ReLU activation functions applied pointwise $\phi(\boldsymbol{x}) = \max(\boldsymbol{x}, \boldsymbol{0})$. Widths of all hidden layers are fixed to the same value $H$. As with all models, all training examples are presented online with batch size 128. Training uses AdamW (Loshchilov and Hutter, 2017) with learning rate $\alpha = 1 \times 10^{-4}$ and weight decay $\lambda = 1 \times 10^{-4}$. The hyperparameters used to train MLPs on each task are presented in Table 1.

Table 1: MLP hyperparameters

| Task | Depth ($\ell$) | Width ($H$) | Train iterations |
|------|------|------|------|
| ICL regression | 2 - 8 | 128 - 2048 | $\leq 2,048,000$ |
| ICL classification | 2 - 8 | 64 - 1024 | $\leq 128,000$ |
| Simple regression | 1 - 4 | 4 - 256 | $\leq 64,000$ |
| Simple classification | 1 - 4 | 4 - 256 | $\leq 64,000$ |
| Match-to-sample | 1 - 4 | 4 - 256 | $\leq 8,000$ |
| Sphere oddball | 1 - 4 | 4 - 256 | $\leq 8,000$ |
| Line oddball | 1 - 4 | 4 - 256 | $\leq 8,000$ |

## C.3 MIXER

The MLP-Mixer accepts inputs $\boldsymbol{X} \in \mathbb{R}^{L \times D}$ (length by token depth). If a task does not provide tokenized inputs, we assume $D = 1$ unless otherwise stated, and reshape accordingly. A model with $\ell$ hidden layers then proceeds as follows:

$$\boldsymbol{h}_1(\boldsymbol{X}) = \phi(\boldsymbol{Z}_1(\boldsymbol{b}_1^\mathsf{T} + \boldsymbol{X}\boldsymbol{W}_1) + \boldsymbol{c}_1)$$
$$\boldsymbol{h}_2(\boldsymbol{X}) = \phi(\boldsymbol{Z}_2(\boldsymbol{b}_2^\mathsf{T} + \boldsymbol{h}_1(\boldsymbol{X})\boldsymbol{W}_2) + \boldsymbol{c}_2)$$
$$\vdots$$
$$\boldsymbol{h}_\ell(\boldsymbol{X}) = \phi(\boldsymbol{Z}_\ell(\boldsymbol{b}_\ell^\mathsf{T} + \boldsymbol{h}_{\ell-1}(\boldsymbol{X})\boldsymbol{W}_\ell) + \boldsymbol{c}_\ell)$$
$$\boldsymbol{f}_{\text{MIX}}(\boldsymbol{X}) = \boldsymbol{W}_{\text{out}}\boldsymbol{h}_\ell(\boldsymbol{X})^{(-1)} + \boldsymbol{b}_{\text{out}}$$

The matrices $\boldsymbol{W}$ mix within token dimensions, and share a fixed hidden width $H$, so $\boldsymbol{W}_i \in \mathbb{R}^{H \times H}$ for $1 < i < \ell$. The matrices $\boldsymbol{Z}$ mix across spatial dimensions, and share a fixed channel width $C$, so $\boldsymbol{Z}_i \in \mathbb{R}^{C \times C}$ for $1 < i < \ell$. The bias vectors $\boldsymbol{b}$ and $\boldsymbol{c}$ are assumed to broadcast over unit dimensions as expected. The index $-1$ in $\boldsymbol{h}_\ell(\boldsymbol{X})^{(-1)}$ refers to taking the last token in the layer, producing an output vector with length $H$. We again use point-wise ReLU activations $\phi(\boldsymbol{X}) = \max(\boldsymbol{X}, 0)$. Our Mixer is a simplified version of the original model proposed in Tolstikhin et al. (2021), and differs in a number of small ways:

- We use only a single hidden layer per Mixer layer, rather than two.

- We apply the point-wise activation *after* the final spatial mixing, and not between spatial and token mixings.

- We do not use layer norm or skip connections.

Using the full original model proved to be unnecessary in our setting, so we proceeded with this simpler version.

As with all models, all training examples are presented online with batch size 128. Training uses AdamW with learning rate $\alpha = 1 \times 10^{-4}$ and weight decay $\lambda = 1 \times 10^{-4}$. The hyperparameters used to train MLPs on each task are presented in Table 2.

Table 2: Mixer hyperparameters

| Task | Depth ($\ell$) | Hidden width ($H$) | Channel width ($C$) | Train iterations |
|------|------|------|------|------|
| ICL regression | 2 - 8 | 32 - 512 | 64 | $\leq 500,000$ |
| ICL classification | 2 - 8 | 16 - 256 | 64 | $\leq 24,000$ |

## C.4 TRANSFORMER

The Transformer accepts inputs $\boldsymbol{X} \in \mathbb{R}^{L \times D}$ (length by token depth). If a task does not provide tokenized inputs, we assume $D = 1$ unless otherwise stated, and reshape accordingly. A model with

$\ell$ hidden layers then proceeds as follows:

$$\tilde{\boldsymbol{X}} = \boldsymbol{X} + PE(\boldsymbol{X})$$

$$\boldsymbol{a}_1(\boldsymbol{X}) = LN(\boldsymbol{A}_1\tilde{\boldsymbol{X}}\boldsymbol{V}_1 + \tilde{\boldsymbol{X}})$$

$$\boldsymbol{h}_1(\boldsymbol{X}) = LN(\boldsymbol{c}_1^{\mathsf{T}} + \phi(\boldsymbol{b}_1^{\mathsf{T}} + \boldsymbol{a}_1(\boldsymbol{X})\boldsymbol{W}_1^{(1)})\boldsymbol{W}_1^{(2)} + \boldsymbol{X})$$

$$\vdots$$

$$\boldsymbol{a}_\ell(\boldsymbol{X}) = LN(\boldsymbol{A}_\ell\boldsymbol{h}_{\ell-1}(\boldsymbol{X})\boldsymbol{V}_\ell + \boldsymbol{X})$$

$$\boldsymbol{h}_\ell(\boldsymbol{X}) = LN(\boldsymbol{c}_\ell^{\mathsf{T}} + \phi(\boldsymbol{b}_\ell^{\mathsf{T}} + \boldsymbol{a}_\ell(\boldsymbol{X})\boldsymbol{W}_\ell^{(1)})\boldsymbol{W}_\ell^{(2)} + \boldsymbol{X})$$

$$\boldsymbol{f}_{\text{TR}}(\boldsymbol{X}) = \boldsymbol{W}_{\text{out}}\boldsymbol{h}_\ell(\boldsymbol{X})^{(-1)} + \boldsymbol{b}_{\text{out}}$$

The attention matrices $A_i$ are single-headed, and constructed as

$$A_i = \sigma\left(\text{mask}\left(\frac{1}{\sqrt{H}}(Q_iX_i)(K_iX_i)^{\mathsf{T}}\right)\right)$$

where "mask" corresponds to a causal attention mask, and $\sigma$ refers to a softmax applied per query. As is now popular, we use GeLU activations applied pointwise for $\phi$. We fix the hidden dimension across all key, query, value, and weight matrices to be of width $H$. We use sinusoidal positional encodings for $PE$ and layer normalization as indicated by $LN$. One exception is for ICL regression, which does not require positional encodings due to the input format (Appendix C.6), so they are omitted in this case. The bias vectors $\boldsymbol{b}$ and $\boldsymbol{c}$ are assumed to broadcast over unit dimensions as expected. The index $-1$ in $\boldsymbol{h}_\ell(\boldsymbol{X})^{(-1)}$ refers to taking the last token in the layer, producing an output vector with length $H$. Our architecture is precisely the decoder-only Transformer architecture first described in Vaswani et al. (2017), with the exception that we do not use dropout.

As with all models, all training examples are presented online with batch size 128. Training uses AdamW with learning rate $\alpha = 1 \times 10^{-4}$ and weight decay $\lambda = 1 \times 10^{-4}$. The hyperparameters used to train MLPs on each task are presented in Table 3.

Table 3: Transformer hyperparameters

| Task | Depth ($\ell$) | Width ($H$) | Train iterations |
|------|------|------|------|
| ICL regression | 2 - 8 | 32 - 512 | $\leq 600,000$ |
| ICL classification | 2 - 8 | 16 - 256 | $\leq 16,000$ |
| Simple regression | 1 - 4 | 8 - 32 | $\leq 256,000$ |
| Simple classification | 1 - 4 | 8 - 32 | $\leq 128,000$ |
| Match-to-sample | 1 - 4 | 8 - 32 | $\leq 8,000$ |
| Sphere oddball | 1 - 4 | 8 - 32 | $\leq 8,000$ |
| Line oddball | 1 - 4 | 8 - 32 | $\leq 8,000$ |

## C.5 RB MLP

The relationally-bottlenecked MLP is architecturally identically to the vanilla MLP described above in Appendix C.2, but with the crucial difference that the inputs are preprocessed to preserve only (dot-product) relations.

The RB MLP accepts inputs $\boldsymbol{X} \in \mathbb{R}^{L \times D}$ (length by token depth). The inputs are processed into a relation matrix $\boldsymbol{R}$ such that each entry is

$$R_{ij} = (\boldsymbol{x}_i - \overline{\boldsymbol{x}}) \cdot (\boldsymbol{x}_j - \overline{\boldsymbol{x}})$$

where $\boldsymbol{x}_i \in \mathbb{R}^D$ refers to the $i$th row of $X$, and $\overline{\boldsymbol{x}} = \frac{1}{L}\sum_i \boldsymbol{x}_i$ is the average across all $\boldsymbol{x}_i$. Relations vectors $\boldsymbol{r}$ are then generated by either selecting a specific column $\boldsymbol{r} = \boldsymbol{R}^{(j)}$ (as in the MTS task) or flattening the entire matrix of relations $\boldsymbol{r} = \text{flat}(\boldsymbol{R})$. The output of the RB MLP is then simply

$$\boldsymbol{f}_{\text{RB}}(\boldsymbol{r}) = \boldsymbol{W}_{\text{out}}\boldsymbol{r} + \boldsymbol{b}_{\text{out}}$$

For the "deep" RB MLP used in the line oddball task, there is an additional set of two hidden layers between $\boldsymbol{r}$ and the readout weights $\boldsymbol{W}_{\text{out}}$, with width 256. All other training parameters are equivalent to the above models.

## C.6 ICL REGRESSION

We prepare in-context regression in a setup that closely mimics Raventós et al. (2024), though without an autoregressive objective. The input consists of a sequence of values $(\boldsymbol{x}_1, y_1), (\boldsymbol{x}_2, y_2), \ldots, (\boldsymbol{x}_L, y_L)$, where $\boldsymbol{x}_i \in \mathbb{R}^n$ and $y_i \in \mathbb{R}$. The $\boldsymbol{x}_i, y_i$ pairs are linearly related through a set of weights $\boldsymbol{\beta} \in \mathbb{R}^n$ such that $y_i = \boldsymbol{x}_i \cdot \boldsymbol{\beta} + \varepsilon$, where $\varepsilon \sim \mathcal{N}(0, \sigma^2)$ corresponds to noise. Finally, the input includes a query $\boldsymbol{x}_q$. The model output is a single scalar regressed against the corresponding $y_q$. Inputs are sampled as $\boldsymbol{x} \sim \mathcal{N}(\boldsymbol{0}, \boldsymbol{I})$ and weights are sampled as $\boldsymbol{\beta} \sim \mathcal{N}(\boldsymbol{0}, \boldsymbol{I}/n)$. Before being presented to the model, all inputs are packed into an input matrix $\tilde{\boldsymbol{X}} \in \mathbb{R}^{(L+1) \times (n+1)}$ with the following structure (Zhang et al., 2023)

$$\tilde{\boldsymbol{X}} = \begin{pmatrix} \boldsymbol{x}_1 & \boldsymbol{x}_2 & \cdots & \boldsymbol{x}_L & \boldsymbol{x}_q \\ y_1 & y_2 & \cdots & y_L & 0 \end{pmatrix}$$

The model returns a scalar value estimate of $y_q$, and is trained using the mean-squared-error. Note: this format does not require positional encodings. Following Zhang et al. (2023), we omit positional encodings for this task.

As in Raventós et al. (2024), we fix a finite pool of weights before training $\boldsymbol{\beta}_1, \boldsymbol{\beta}_2, \ldots, \boldsymbol{\beta}_k$, where $\boldsymbol{\beta}_i \sim \mathcal{N}(\boldsymbol{0}, \boldsymbol{I}/n)$. For each training example, we sample a new $\boldsymbol{\beta}$ by selecting uniformly at random one weight from the pool $\{\boldsymbol{\beta_i}\}_{i=1}^k$. We also consider the limit $k \to \infty$, which corresponds to sampling $\boldsymbol{\beta} \sim \mathcal{N}(\boldsymbol{0}, \boldsymbol{I}/n)$ afresh rather than drawing from a fixed pool. During testing, we probe the model's performance both on the training distribution where the weights are restricted to a finite pool $\boldsymbol{\beta} \sim \mathcal{U}\left(\{\boldsymbol{\beta_i}\}_{i=1}^k\right)$ and an unrestricted distribution where the weights are drawn freely $\boldsymbol{\beta} \sim \mathcal{N}(\boldsymbol{0}, \boldsymbol{I}/n)$.

Unless stated otherwise, all of our experiments use $n = 8$ dimensional regression with $L = 8$ points in the context, and noise level $\sigma^2 = 0.05$.

**Bayes estimators.** We compare our models to two different Bayes estimators that correspond to priors assuming finite or infinite $k$.

For finite $k$ where weights $\boldsymbol{\beta}$ are sampled uniformly from a pool of $k$ possibilities, the Bayes optimal estimator is given by the discrete minimum mean-squared error (dMMSE) estimator, based on the estimator formulated in Raventós et al. (2024)

$$\hat{\boldsymbol{\beta}}_{\text{dMMSE}} = \sum_{i=1}^k w_i \boldsymbol{\beta}_i$$

where the weights $w_i$ are given by

$$w_i \propto \exp\left\{-\frac{1}{2\sigma^2} \sum_{j=1}^L (y_j - \boldsymbol{x}_j \cdot \boldsymbol{\beta}_i)^2\right\}$$

normalized such that $\sum_i w_i = 1$.

In the case $k \to \infty$, the Bayes optimal estimator is simply the familiar Ridge estimator with Bayes optimal regularization

$$\hat{\boldsymbol{\beta}}_{\text{Ridge}} = \left(\boldsymbol{X}^\intercal \boldsymbol{X} + n\sigma^2 \boldsymbol{I}\right)^{-1} \boldsymbol{X}^\intercal \boldsymbol{y}$$

where the rows of $\boldsymbol{X}$ are the context points, and $\boldsymbol{y} = (y_1, y_2, \ldots, y_L)$ are the corresponding labels.

## C.7 ICL CLASSIFICATION

We prepare ICL classification in a setup that closely mimics Reddy (2024). We begin with a set of labels $\boldsymbol{\alpha}_1, \boldsymbol{\alpha}_2, \ldots \boldsymbol{\alpha}_C \in \mathbb{R}^n$ that correspond to class indices $1, 2, \ldots C$. Labels are sampled as $\alpha \sim \mathcal{N}(\boldsymbol{0}, \boldsymbol{I}/n)$. The model ultimately predicts the class index, but the real-valued labels provide content of the correct dimension to fill an input without arbitrary padding (described further below).

Points are sampled from a Gaussian mixture model $\mathcal{M}_k$ consisting of $k$ components, where $k \geq C$ (we allow multiple clusters to have the same class label). Each component is associated with a center

$\boldsymbol{\mu}_k \sim \mathcal{N}(\mathbf{0}, \boldsymbol{I}/n)$. A point is sampled from the $k$th component as

$$\boldsymbol{x}_k = \frac{\boldsymbol{\mu}_k + \varepsilon \boldsymbol{\eta}}{\sqrt{1 + \varepsilon^2}}$$

where $\boldsymbol{\eta} \sim \mathcal{N}(\mathbf{0}, \boldsymbol{I}/n)$ and $\varepsilon$ governs the within-cluster variability. Below in Figure 8, we also consider a $k \to \infty$ setting, where the number of mixture components is infinite. This settings corresponds to a case where the mixture centers $\boldsymbol{\mu}_k$ are resampled for each example, always producing novel clusters. In the finite $k$ case, mixture centers remain fixed throughout the duration of the task.

An input sequence consists of $L$ context exemplars $(\boldsymbol{x}_1, \boldsymbol{y}_1), (\boldsymbol{x}_2, \boldsymbol{y}_2), \dots, (\boldsymbol{x}_L, \boldsymbol{y}_L)$ followed by a query point $\boldsymbol{x}_q$, where $\boldsymbol{x}_i \sim \mathcal{M}_k$ and $\boldsymbol{y}_i \in \{\boldsymbol{\alpha}_j\}$ is the corresponding label for the cluster that originated the point. The model must predict the corresponding query label $\boldsymbol{y}_q$, and output the class index associated with this label. The inputs are packed into an input matrix $\tilde{\boldsymbol{X}} \in \mathbb{R}^{(2L+1) \times n}$ which has structure

$$\tilde{\boldsymbol{X}} = (\boldsymbol{x}_1 \quad \boldsymbol{y}_1 \quad \boldsymbol{x}_2 \quad \boldsymbol{y}_2 \quad \cdots \quad \boldsymbol{x}_L \quad \boldsymbol{y}_L \quad \boldsymbol{x}_q)$$

The model outputs logits over class indices, and is trained using cross-entropy loss.

We also parameterize the inputs by *burstiness* $B$, which is the number of repeats per cluster in the context ($B$ must divide the context length $L$). For example, $B = 2$ means there are exactly two points from each cluster represented in the inputs.

Unless otherwise stated, we use $n = 8$ dimensional inputs, $C = 32$ class labels, and within-cluster variability $\varepsilon = 0.1$.

## C.8 MATCH-TO-SAMPLE

The match-to-sample task proceeds as follows. The model is presented with $L$ context points $\boldsymbol{x}_1, \boldsymbol{x}_2, \dots, \boldsymbol{x}_L \in \mathbb{R}^n$ followed by a query point $\boldsymbol{x}_q$. The inputs are packed into an input matrix $\tilde{\boldsymbol{X}} = (\boldsymbol{x}_1, \boldsymbol{x}_2, \dots, \boldsymbol{x}_L, \boldsymbol{x}_q) \in \mathbb{R}^{(L+1) \times n}$ before being passed to the model.

The context points are evenly distributed along a sphere $\mathcal{S}^n$ with unit radius centered at the origin. Points are rotated by a random angle so that their absolute positions vary from input to input. The model must return the index of the context point closest to the query $y = \arg\max_i (\boldsymbol{x}_i \cdot \boldsymbol{x}_q)$, and is trained using cross-entropy loss.

Unless otherwise stated, we use $L = 5$ context points and $n = 2$ dimensional inputs.

## C.9 SPHERE ODDBALL

The sphere oddball task proceeds as follows. The model is presented with $L$ context points $\boldsymbol{x}_1, \boldsymbol{x}_2, \dots, \boldsymbol{x}_L \in \mathbb{R}^n$. (There are no query points.) The context points are sampled as $\boldsymbol{x} \sim \mathcal{N}(\boldsymbol{\mu}, \boldsymbol{I})$. The center is sampled uniformly from a box $\boldsymbol{\mu} \sim \mathcal{U}[-B, B]^n$. One point in the context is selected at random and perturbed in a random direction $\boldsymbol{v}$ with magnitude $d = ||\boldsymbol{v}||$, so that $\boldsymbol{x}_{\text{oddball}} \leftarrow \boldsymbol{x}_{\text{oddball}} + \boldsymbol{v}$. The model must return the index $y$ of the oddball point in the context, and is trained using cross-entropy loss. Both the center $\boldsymbol{\mu}$ and points $\boldsymbol{x}_i$ are sampled afresh from example to example, necessitating a general relational solution.

Unless otherwise stated, we use $n = 2$ dimensional inputs, $L = 6$ points in the context, and a box size of $B = 10$.

## C.10 LINE ODDBALL

The line oddball task proceeds as follows. For each training example, we select an $n - 1$ dimensional plane with random orientation that passes through the origin. Context points $\boldsymbol{x}_1, \boldsymbol{x}_2, \dots, \boldsymbol{x}_L \in \mathbb{R}^n$ are Gaussian distributed along this subspace with zero mean and unit variance. One context point is selected at random to be the oddball, and is perturbed by a distance $d$ in the direction perpendicular to the line. The model must output the index $y$ of the oddball point, and is trained using cross-entropy.

Unless otherwise stated, we use $n = 2$ dimensional inputs and $L = 6$ points in the context.

### C.11 SIMPLE REGRESSION

Simple (non-ICL) regression is the classic regression setup. The model receives as input a single point $\boldsymbol{x} \in \mathbb{R}^n$, and must output the corresponding $y \in R$ which is related through $y = \boldsymbol{x} \cdot \boldsymbol{\beta} + \varepsilon$. Weights are sampled as $\boldsymbol{\beta} \sim \mathcal{N}(\boldsymbol{0}, \boldsymbol{I}/n)$, and noise is sampled as $\varepsilon \sim \mathcal{N}(0, \sigma^2)$. Weights $\boldsymbol{\beta}$ are sampled once, then remain fixed through the entire duration of the task. The model is trained using MSE loss.

Unless otherwise stated, we consider $n = 64$ dimensional regression with noise level $\sigma^2 = 0.05$.

In Appendix D, we also consider a simple non-linear version of regression where $y = (\boldsymbol{x} \cdot \boldsymbol{\beta})^p + \varepsilon$ for powers $p = 2$ and $3$.

### C.12 SIMPLE CLASSIFICATION

Simple (non-ICL) classification proceeds as follows. The model receives as input a single point $\boldsymbol{x} \in \mathbb{R}^n$ that we sample from 1 of $k$ different clusters. Cluster centers $\boldsymbol{\mu}_i$ are sampled as $\boldsymbol{\mu}_i \sim \mathcal{N}(\boldsymbol{0}, \boldsymbol{I}/n)$. The label $y$ of $\boldsymbol{x}$ is given by

$$y = \arg\min_i ||\boldsymbol{x} - \boldsymbol{\mu}_i||$$

Cluster centers are sampled once, then remain fixed throughout the entire duration of the task. The model is trained using cross-entropy loss.

Unless otherwise stated, we consider $n = 64$ dimensional inputs with $k = 16$ classes.

### C.13 COMPUTE

To measure the number of floating point operations (FLOPs) used to train a model, we use Jax's cost analysis routines. Specifically, we compute the total number of FLOPs required to perform a single step of gradient descent, then multiply this quantity by the total number of gradient steps used to train the model.

All experiment were run on the Harvard FASRC research cluster. CPU requirements are negligible compared to GPU time, so they are omitted. All experiments required no more than 16 GB of RAM. The per-experiment GPU time on an A100 to generate the above figures are estimated at

- **ICL regression**: 1500 GPU hours
- **ICL classification**: 500 GPU hours
- **Simple regression**: 50 GPU hours
- **Simple classification**: 50 GPU hours
- **Match-to-sample**: 10 GPU hours
- **Sphere oddball**: 10 GPU hours
- **Line oddball**: 10 GPU hours

The total GPU time is therefore roughly 2130 GPU hours. The compute used to generate these results represents less than 5 percent of the total compute deployed through the life-cycle of this research project.

### C.14 SOFTWARE

All models are implemented and trained using the Jax (Bradbury et al., 2018) family of libraries, particularly Flax (Heek et al., 2023). Plots are created using Seaborn (Waskom, 2021) and Pandas (pandas development team, 2020).

## D    ADDITIONAL FIGURES

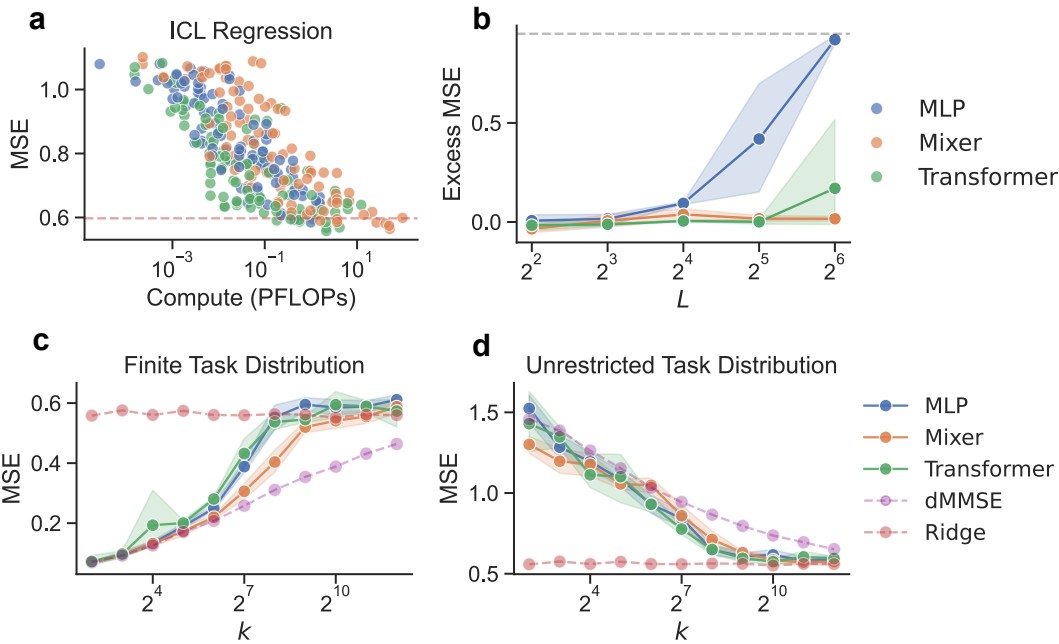

Figure 5: **ICL regression with an autoregressive objective.** For each input example $(\boldsymbol{x}_1, y_1, \boldsymbol{x}_2, y_2, \ldots, \boldsymbol{x}_L, y_L)$, we compute the autoregressive loss $\sum_i \mathcal{L}(f(\boldsymbol{x}_1, y_1, \boldsymbol{x}_2, y_2, \ldots \boldsymbol{x}_i), y_i)$, for a neural network $f$ and MSE loss $\mathcal{L}$. For vanilla MLPs and Mixers, variable-length inputs are handled by padding inputs with zero up to the max length $L$. **(a)** Compute vs. MSE on the unrestricted task distribution. Each point represents a single model, with particular parameters and training iterations. Just as in the fixed input length case, at large compute, MSE is approximately equal across all architectures. The red line corresponds to the Bayes optimal Ridge MSE. **(b)** Excess MSE (MSE above Bayes optimal) for varying context length $L$ on the unrestricted task distribution. Excess MSE remains flat for Mixers and Transformers, but rises for MLPs. The grey line corresponds to the excess MSE incurred by the zero predictor. Given compute limitations, we plot on a slightly narrower range of context lengths, but the overall trends remain consistent with the finite-input-length case. **(c, d)** IWL to ICL transition with increasing data diversity. We train on a finite distribution with $k$ weights, then test on both the finite training distribution and the unrestricted distribution. Just as with finite input lengths, all models exhibit a transition from IWL (represented by dMMSE) to ICL (represented by Ridge) as $k$ increases. Note: it is possible to "outperform" Bayes optimal Ridge on the finite training distribution by learning in-weight the underlying $\boldsymbol{\beta}$'s. **(all)** We use $n = 8$ dimension inputs. All line plots feature 95 percent confidence intervals about the mean, estimated from 5 replications.

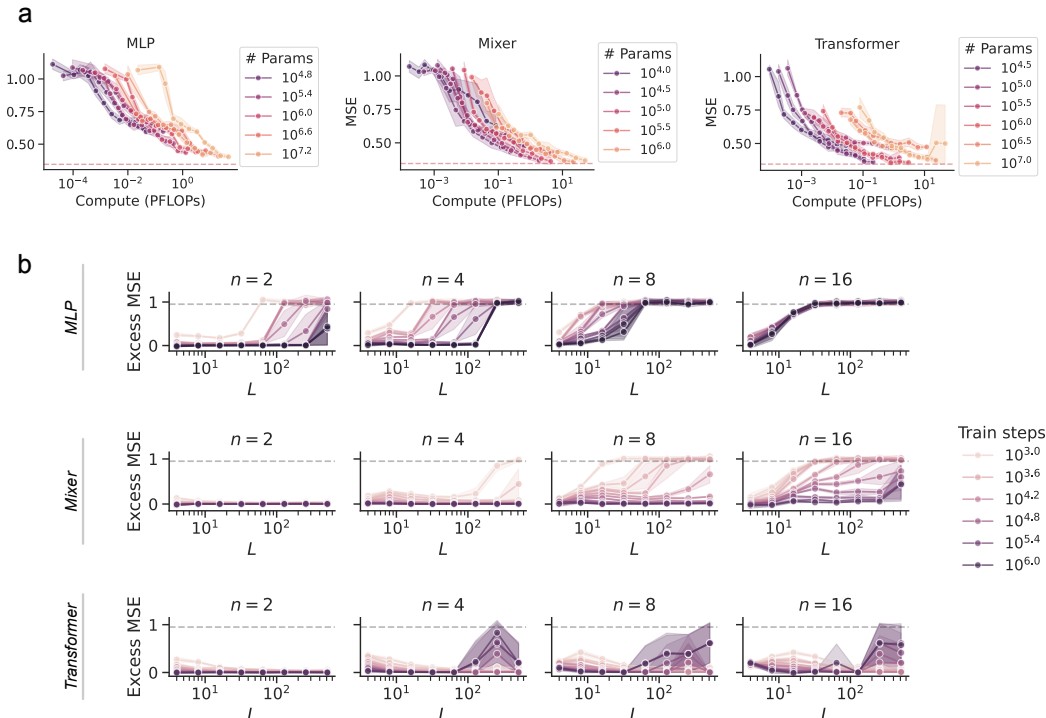

Figure 6: **ICL regression supplementary figures. (a)** MSE obtained across each architecture as a function of compute. Lines connect common models, with colors denoting different parameter counts. Hence, a single line traces the trajectory of a model across different training iterations. The red dashed line corresponds to the Bayes optimal MSE. **(b)** Excess MSE across different context lengths $L$, for different input dimensions $n$. Line colors indicate the number of elapsed training steps. The gray dashed line corresponds to the MSE obtained from always guessing zero. Particularly for high dimensions, MLPs struggle to learn in-context with larger context lengths. After sufficient training, both Mixers and Transformers can learn in-context even for very large input contexts. **(all)** Shaded regions correspond to 95 percent confidence intervals computed across 5 replications.

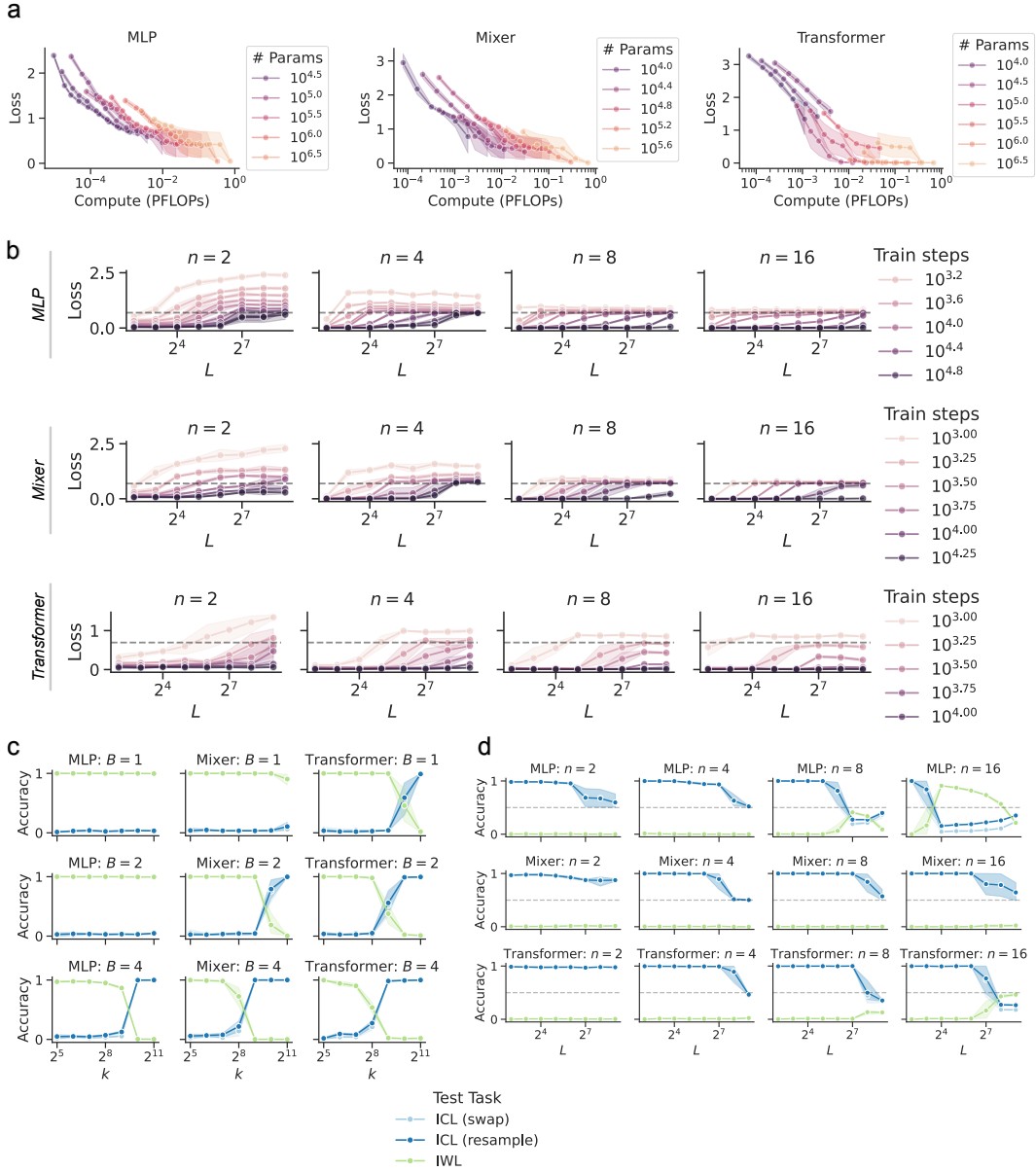

Figure 7: **ICL classification supplementary figures.** **(a)** MSE obtained across each architecture as a function of compute. Lines connect common models, with colors denoting different parameter counts. Hence, a single line traces the trajectory of a model across different training iterations. **(b)** Cross entropy loss across different context lengths $L$, for different input dimensions $n$. Line colors indicate the number of elapsed training steps. In these examples, $B = n/2$, so there are only 2 labels present in each context (out of $C = 32$ total possible labels). The gray dashed line corresponds to the loss obtained by placing equal probability on the 2 labels present in the context. All models plateau for a time at guessing one among the two correct labels, before eventually collapsing to the correct ICL solution. **(c)** IWL to ICL transition for different burstiness $B$. Consistent with prior work (Reddy, 2024; Chan et al., 2022), higher burstiness encourages ICL. Transformers transition to ICL for lower burstiness and lower number of clusters $k$. **(d)** ICL vs. IWL behavior for $B = n/2$ and $k = 2048$ clusters across context lengths $L$ and input dimensions $n$. For the most part, these settings are sufficient to encourage ICL, including the configuration plotted in the main text Figure 1, though ICL appears to decay at higher dimensions and longer contexts. **(all)** Line plots feature 95 percent confidence intervals about the mean, computed across 5 replications.

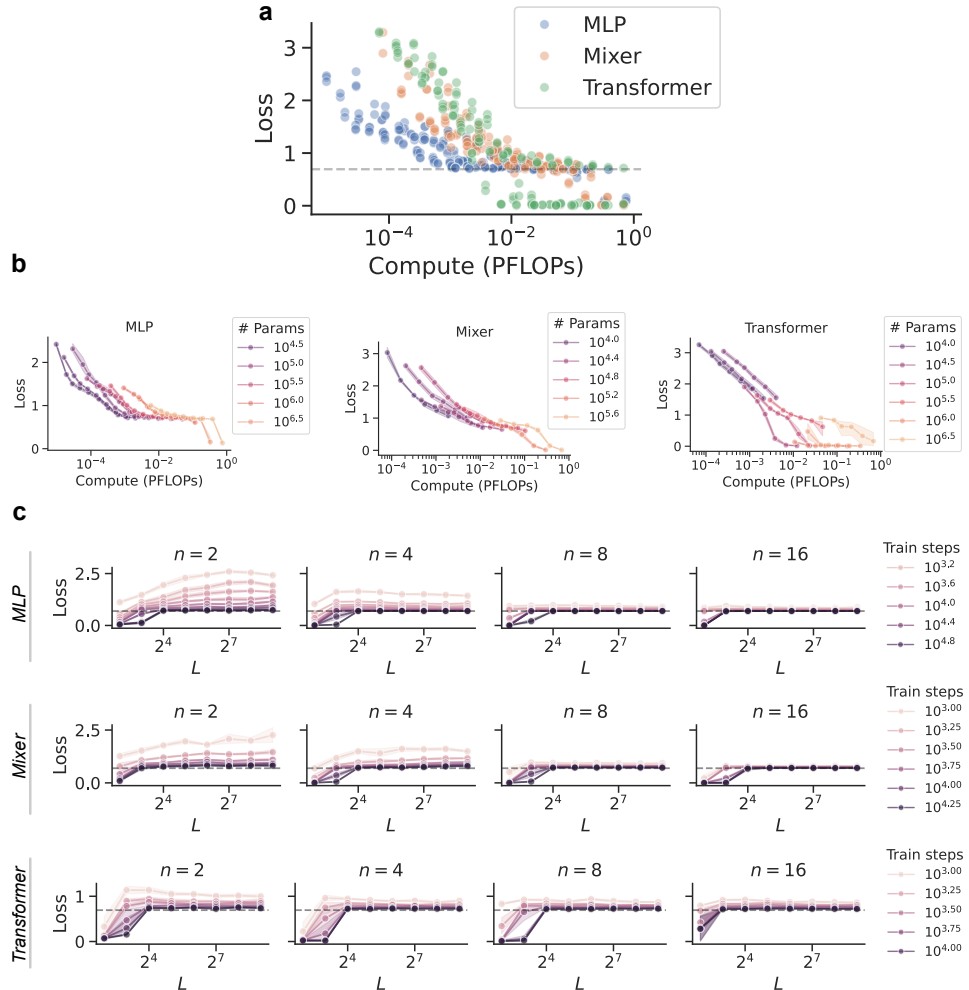

Figure 8: **ICL classification with infinite clusters.** Just as we can consider a $k \to \infty$ limit for ICL regression, where regression weights are sampled afresh for each example, we can consider an analogous $k \to \infty$ limit for ICL classification where clusters are resampled for each new example rather than being fixed to an underlying Gaussian mixture. Doing so forces each model to learn an in-context solution, but the learning outcomes turn out to be different. In particular, the task because substantially more difficult for longer contexts. For example, selecting a context length $L = 16$ with infinite clusters is enough to block any model from learning the full ICL solution. In contrast, $L = 16$ with finite clusters can still push a model to learn the full ICL solution (Figure 7), even if an in-weight solution is also available. For this reason, we consider only finite but large $k$ in the main text, enough to develop ICL without blocking learning for longer contexts. In this appendix figure, we examine more closely what happens if we attempt ICL classification with infinite clusters. **(a)** Loss obtained by each architecture as a function of compute, for context length $L = 8$ and $n = 8$ dimensional inputs with burstiness $B = 4$, so 2 of the 32 possible labels appears in each example. The gray dashed line corresponds to the loss obtained by a model if it assigns equal probability to the 2 labels present in the example. Like in Figure 1, we witness a plateau at the gray line, though it is somewhat more severe. Nonetheless, all models are able to perform the task perfectly with sufficient compute. **(b)** Line plot for each architecture in panel (a). Lines connect common models, with colors denoting different parameter counts. Hence, a single line traces the trajectory of a model across different training iterations. **(c)** Cross entropy loss across different context lengths $L$, for different input dimensions $n$. Line colors indicate the number of elapsed training steps. The gray dashed line corresponds to the loss obtained by placing equal probability on the 2 labels present in the context among the 32 total labels. For context lengths $L \geq 16$, all models plateau at the gray line and fail to learn further. Hence, it appears that even Transformers fail to learn the full in-context task, and remain stuck at a local optima of guessing one of the two labels present in the context. In contrast, if we fixed the number of clusters $k$ to a large but finite value, all models will learn the full ICL solution even though an in-weight solution is available (Figure 7 above). In this way, it appears that finite clusters afford some curricular benefit that leads a model to the ICL solution, which the infinite case lacks. This discrepancy poses a fascinating topic for future study. **(all)** Shaded regions correspond to 95 percent confidence intervals computed across 5 replications.

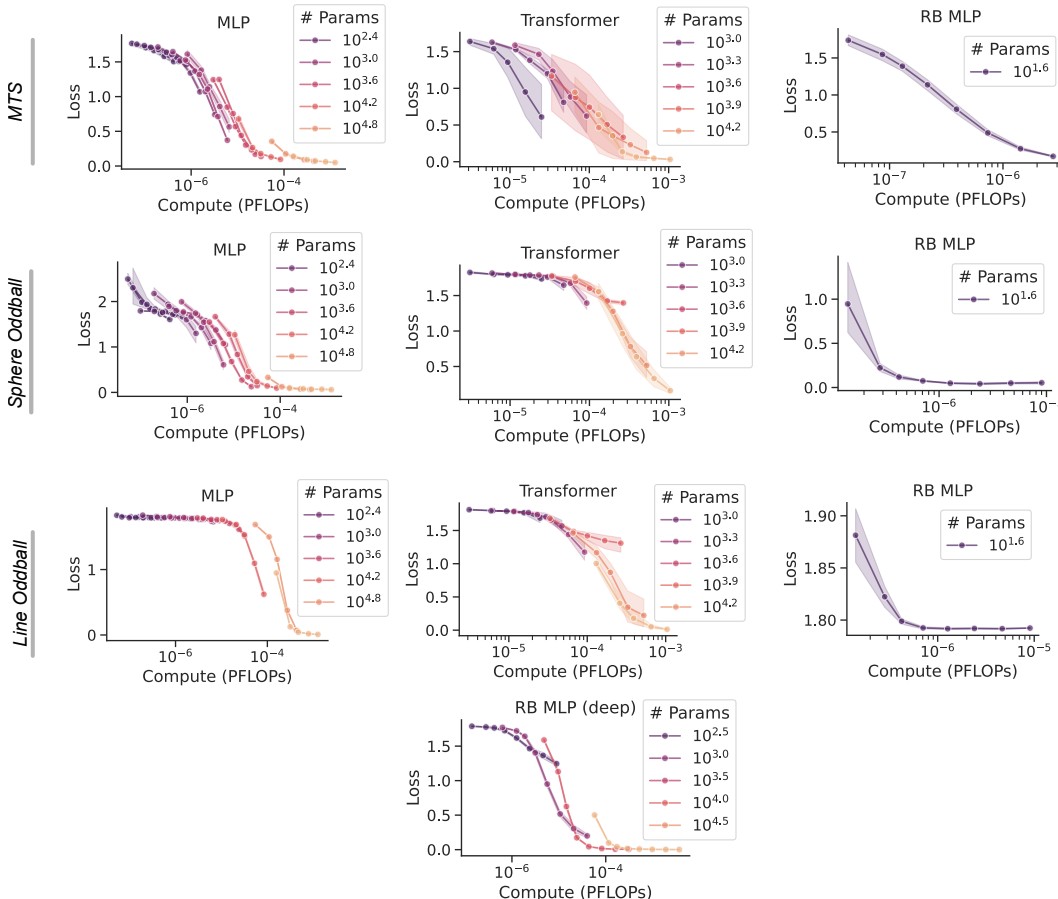

Figure 9: **Relational reasoning supplementary figures.** We plot the loss obtained across each architecture as a function of compute. Lines connect common models, with colors denoting different parameter counts. Hence, a single line traces the trajectory of a model across different training iterations. Note: the RB MLP does not have configurable widths or depths, so all RB MLPs have the same parameter count. **(all)** Shaded regions correspond to 95 percent confidence intervals computed across 5 replications.

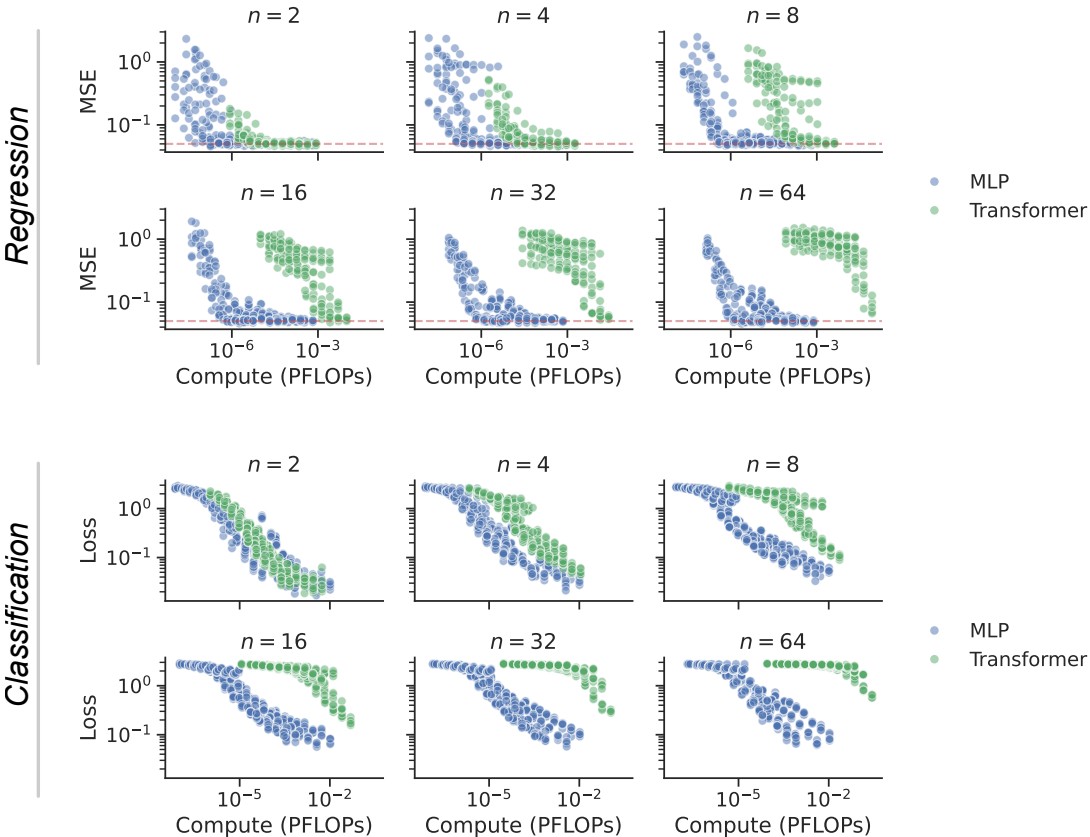

Figure 10: **Simple regression and classification with varying input dimension.** We plot the MSE (for regression) or cross entropy loss (for classification) as a function of compute across varying input dimension $n$. The red dashed lines in the regression plots correspond to the minimum attainable MSE. Each point corresponds to a single model with a particular parameter and training time. In all cases, reducing the dimension of the input reduces the gap between Transformers and MLPs, with the gap effectively vanishing for $n = 2$ dimensional inputs.

