# OpenReview forum: "MLPs Learn In-Context on Regression and Classification Tasks"
_ICLR.cc/2025/Conference — ICLR 2025 Poster_

### Official Review · Reviewer_3xiz · 2024-10-21

**Soundness:** 2
**Presentation:** 3
**Contribution:** 1
**Rating:** 3
**Confidence:** 4

**Summary:**

This paper investigates the in-context learning ability of MLP and MLP-Mixer, and find it learns in-context competitively with Transformers given the same compute budget in some simple settings.

**Strengths:**

This paper is well written. The connection with related works are clearly presented.

**Weaknesses:**

The contribution of this work is hard to identify, and the results are not novel nor inspiring. First, it has been widely studied that MLP can learn relational data, i.e., can predict for set/sequence inputs, i.e., can learn in-context. Thus the studied topic is not novel. Second, it is also known that naive MLP for relational prediction (concatenating input) has disadvantages, e.g. lack of capturing permutation-invariance or variant input number. But this paper does not point out this keys, which would be helpful to explain why the MLP-Mixer and RB MLP can learn in-context better. Third, the result that MLP competitively with Transformers given the same compute budget is obtained with a synthetic setting, considering above nature of MLPs, it is not convincing enough to be generalizable.

In playful but intuitive words, while RB MLP and MLP-Mixer can be viewed as MLP Pro for relational data (including ICL), Transformer can be viewed as MLP Pro Max. So while it is mainstream to study ICL with transformer, it is not meaningful enough to study ICL with MLP.

**Questions:**

Could the authors provide empirical results on real-world few-shot learning data?

---

> ### Author Response · Authors · 2024-11-15
>
> Thanks for the comments! We’re glad you found the paper to be well-written and clear. Answering your questions:
>
> >The contribution of this work is hard to identify, and the results are not novel nor inspiring. First, it has been widely studied that MLP can learn relational data, i.e., can predict for set/sequence inputs, i.e., can learn in-context. Thus the studied topic is not novel.
>
> We’re sorry to hear your thoughts, and will strive to clarify our contributions in our manuscript. We are unaware of prior work that has demonstrated MLPs can learn in-context, but would be very interested to know what references you had in mind? There is certainly related work we cite that rely on architectural modifications like relational bottlenecks or graph neural networks purpose-built for relational tasks, but none as far as we are aware that examine unmodified MLPs or Mixer models. Indeed, there is active literature arguing that vanilla MLPs *cannot* learn in-context on relational tasks ([Boix-Adsera et al](https://arxiv.org/abs/2310.09753) make an especially strong claim in this regard. See also [Webb et al](https://arxiv.org/abs/2309.06629), [Santoro et al](https://arxiv.org/abs/1706.01427), and [Marcus](https://www.sciencedirect.com/science/article/pii/S0010028598906946)), which we refute in our manuscript.
>
>
> >Second, it is also known that naive MLP for relational prediction (concatenating input) has disadvantages, e.g. lack of capturing permutation-invariance or variant input number. But this paper does not point out this keys, which would be helpful to explain why the MLP-Mixer and RB MLP can learn in-context better.
>
> Thanks for the note! It is indeed surprising that the disadvantaged, naive MLP outperforms the Transformer on our relational tasks, despite the weaknesses you mention. Documenting surprising performance gaps like these is a core contribution of our work. We will emphasize these points further in our manuscript.
>
> >Third, the result that MLP competitively with Transformers given the same compute budget is obtained with a synthetic setting, considering above nature of MLPs, it is not convincing enough to be generalizable.
>
> Great question. We were interested in cleanly evaluating ICL performance alone, and sought to avoid confounds introduced in naturalistic settings like developing fluency in natural language or learning complex visual features (which are certainly important topics of study in their own right, but outside the scope of our investigation). As a result, we drew from simple, synthetic ICL tasks common in the literature. We do cite other work demonstrating that MLP Mixers in particular continue to perform competitively with Transformers in real-world NLP and computer vision settings (for example [Liu et al](https://arxiv.org/abs/2105.08050), [Tolstikhin et al](https://arxiv.org/abs/2105.01601)), suggesting that our conclusions may continue to hold in these cases as well.
>
> >In playful but intuitive words, while RB MLP and MLP-Mixer can be viewed as MLP Pro for relational data (including ICL), Transformer can be viewed as MLP Pro Max. So while it is mainstream to study ICL with transformer, it is not meaningful enough to study ICL with MLP.
>
> We appreciate the fun and quirky analogy! To clarify, we are not proposing that Transformers should be replaced wholesale by MLPs. Rather, MLPs are an important class of neural networks that are still intensely studied in the present day, with hitherto untested capabilities on in-context learning. That ICL can be accomplished by such a simple model was very surprising to us, and runs against the conventional wisdom that ICL is uniquely suited for attention-based architectures. To improve our understanding of this phenomenon, it is very worthwhile documenting the properties of these models on commonly studied in-context learning tasks.
>
> >Could the authors provide empirical results on real-world few-shot learning data?
>
> Thanks for the suggestion! As mentioned above, our focus has deliberately been on a synthetic setting where the models’ specific competency for ICL can be clearly interrogated, without the confounds introduced by real-world data. Naturalistic data falls outside the scope of this manuscript, but we look forward to investigating it further in the future.

---

> > ### Comment · Reviewer_3xiz · 2024-11-22
> > **Reply**
> >
> > Thanks for the authors' feedback.
> >
> > Naive MLPs are not equipped with certain important desired properties to be a sequence/set encoder to perform ICL, e.g., the capacity of handling variant length input, permutation-invariant. The reviewer is not surprised that MLP can ICL, but not convinced that they can ICL well on more general settings.
> >
> > To address above limitations and generalize the input of DNN to data with set/sequence structures, people have worked in decades to design more feasible architectures from introducing pooling layers, to RNNs, contemporary transformers and SSMs. I personally think that going back to the start is not meaningful enough, if no further insights can be provided.
> >
> > I keep my negative opinion of this paper unchanged.

---

> > > ### Author Response · Authors · 2024-11-22
> > >
> > > We’re sorry to hear your final thoughts. Note, we never advocate the wholesale replacement of other architectures with MLPs, and certainly support ongoing work examining architectural advancements in neural networks.
> > >
> > > More broadly, surprise is a subjective feeling, and others in our community have different priors and were quite surprised by our findings. There’s merit in scientific study, even if (or perhaps especially if) investigators have strong priors that can be falsified. Certainly, it would be unsurprising if MLPs could *not* perform ICL, given their severe limitations as you listed. That MLPs *can* perform ICL, despite their simplicity, was quite surprising to us, and tempers the great deal of theoretical focus on the role of architecture (particularly Transformers) in ICL. The intent is not to replace Transformers, but to broaden our scientific understanding of ICL and MLPs.
> > >
> > > Thanks for the discussion!

---

### Official Review · Reviewer_v7Gh · 2024-10-28

**Soundness:** 4
**Presentation:** 4
**Contribution:** 3
**Rating:** 6
**Confidence:** 3

**Summary:**

The authors empirically study the performance of MLPs, MLP Mixer Models, Transformers and some hand crafted models on a range of small synthetic in-context learning tasks. By carefully designing the tasks the authors show a transition from in "weight learning" to in "context learning (ICL)".

**Strengths:**

The paper is thought provoking, well written and does a good job a questioning common assumptions, such as which Models can exhibit ICL? and what ICL even is?

**Weaknesses:**

The paper compares transformers applied auto-regressively against MLP and MLP mixer models that access the data all at once (are not auto-regressive). This seems like a strange choice for a fair comparison especially when trying to compare training FLOPs required. I strongly suggest including a bi-directional transformer as a fairer baseline. This would greatly strengthen any claim made about computational efficiency.

The paper does not really offer a concrete definition of what ICL is. “In-context learning (ICL) refers to a task paradigm where exemplars from a novel task are presented during inference time rather than during training. ” One could argue working out a linear projection for a “new point” is a new task and hence standard linear regression is “in-context learning”. For clarity I think promoting this sort of discussion about what is and isn’t ICL is a strength of the paper, but I would like to the see the authors add more discussion on this.

The paper only consider small toy problems which makes it hard to know if these observation generalise to more real world problems.

Similar to the above two points I'm fairly confident some researchers might not consider the problems tackled in the paper "real in-context learning", due to the scale and high levels of structure. I would like to see the authors add more discussion and try and push for a more concrete definition of ICL. The fact that MLP can learn a mapping that corresponds to the linear regression algorithm (or similar) is not that surprising. In my opinion the strength of the paper is questioning the existing definition of ICL and trying to push for a more rigours definition, I would like to see the authors lean into this a bit more.

**Questions:**

Did you try comparing against a bi-directional transformer?

Did you try applying the MLP or MLP mixer auto-regressively?

If you had to give a concrete definition of ICL what would it be?

Why is working out a linear projection for a “new point” not a new task (and not ICL) where as working out the projection of a point in a new linear regression problem ICL? To me the only different seem to be be how you define a "task"?

Did you consider trying to get even more "meta" so each X,Y pair represents a type of regression problem? (say polynomials of a different order, or a mix of classification and regression problems). Each X would represent the concatenation of m examples of a different problems of the same type. The task is to in-context learn the class of the problem then work out the solution to a new point according to the class of problem? In other words apply the construction figure 1, recursively so X=(((x111,x112,...x11d,y11),....,(x1n1,x1n2,...x1nd,y1n)),....,((xm11,xm12,...xm1d,ym1),....,(xmn1,xmn2,...xmnd,ymn))(xqq1,xqq2,...xqqd)) Y = yqq. here you have m examples of problems from the same class each with n points with dimension d.

---

> ### Author Response · Authors · 2024-11-15
> **Discussion: Part I**
>
> Thank you for the equally fascinating and thought-provoking comments! We’re glad you enjoyed reading the paper, and found it well-written and compelling.
>
> The character limitations in openreview prevent us from fully engaging in your discussion points within a single comment box, so we've split them in two. Apologies for the inconvenience! Part I below:
>
> > The paper compares transformers applied auto-regressively against MLP and MLP mixer models that access the data all at once. This seems like a strange choice for a fair comparison especially when trying to compare training FLOPs. I strongly suggest including a bi-directional transformer as a fairer baseline.
>
> Our apologies, we do indeed run experiments in an auto-regressive setting (Fig. 5), where we demonstrate that the exact same results continue to hold. In this case, the MLP and Mixer models are also prompted auto-regressively, with zero-padding  up to max length to handle variable-length inputs. The reference to this figure was easy to miss, so we’ll clarify our manuscript to emphasize its existence. Note: in the non-autoregressive setting, the Transformer is not prompted auto-regressively, so it too sees the entire input before being asked for an output, just like the MLP and Mixer. Incorporating bi-directionality in this case may not produce a difference.
>
> > The paper does not really offer a concrete definition of what ICL is. “In-context learning (ICL) refers to a task paradigm where exemplars from a novel task are presented during inference time rather than during training. ” One could argue working out a linear projection for a “new point” is a new task and hence standard linear regression is “in-context learning”. For clarity I think promoting this sort of discussion about what is and isn’t ICL is a strength of the paper, but I would like to the see the authors add more discussion on this.
>
> Interesting take! We implicitly follow the definition of ICL commonly used in the literature in these synthetic settings (for example, in [Garg et al](https://arxiv.org/abs/2208.01066), [Zhang et al](https://arxiv.org/abs/2306.09927)), where a “task” is defined to be a member of a function class $\mathcal{F}$ such that for every task/function $f \in \mathcal{F}$, the model can correctly predict the response to an input $x_1, f(x_1), x_2, f(x_2), \ldots, x_q$. Under this definition, an experiment consisting of a single linear regression example would constitute a function class $\mathcal{F}$ that contains a single member. While one could conceivably label this as “in-context learning,” it is a bit pathological as in-context learning typically implies that the model is generalizing to unseen, novel functions in $\mathcal{F}$, which we stipulate for our experiments. We will certainly add more discussion on this topic.
>
> >The paper only consider small toy problems which makes it hard to know if these observation generalise to more real world problems.
>
> Great point. We were interested in cleanly evaluating ICL performance, and sought to avoid confounds introduced in naturalistic settings like developing fluency in natural language or learning complex visual features (which are certainly important topics of study in their own right, but outside the scope of our investigation). Hence, we drew from simple, synthetic ICL tasks common in the literature. It is an important direction of future research to see how these conclusions may break down as more complexity is introduced to the task.

---

> > ### Author Response · Authors · 2024-11-15
> > **Discussion: Part II**
> >
> > >Similar to the above two points I'm fairly confident some researchers might not consider the problems tackled in the paper "real in-context learning", due to the scale and high levels of structure. I would like to see the authors add more discussion and try and push for a more concrete definition of ICL. The fact that MLP can learn a mapping that corresponds to the linear regression algorithm (or similar) is not that surprising. In my opinion the strength of the paper is questioning the existing definition of ICL and trying to push for a more rigours definition
> >
> > Thanks for the comment! We absolutely agree that what constitutes a valid task or function class under an ICL setting is a very interesting topic of study in its own right. In our case, we accepted the common definition for synthetic ICL tasks highlighted above, and pursued tasks commonly accepted and studied in this literature ([regression](https://arxiv.org/abs/2306.15063), [classification](https://arxiv.org/abs/2312.03002), [relational tasks](https://arxiv.org/abs/2309.17363)). Please see also our answers below.
> >
> > >Did you try comparing against a bi-directional transformer?
> > Did you try applying the MLP or MLP mixer auto-regressively?
> >
> > Great questions. As noted above, we perform experiments in an auto-regressive setting (Fig. 5), and will clarify our manuscript to make them more visible. In the non-autoregressive setting, our Transformer is not prompted auto-regressively, so it sees the entire input before being asked to respond, precisely like the MLP and Mixer. In this case, adding bi-directionality may not produce a difference.
> >
> > >If you had to give a concrete definition of ICL what would it be?
> >
> > Great question! As mentioned above, the definition from [Garg et al](https://arxiv.org/abs/2208.01066) seems like a great place to start. Perhaps a good way to extend it would be to formally stipulate that the model must correctly generalize to unseen, novel functions in the function class $\mathcal{F}$ during testing. This definition excludes pathological function classes where $|\mathcal{F}| = 1$, since it’s impossible to partition this function class into seen and unseen sets. In this way, classical linear regression is not in-context learning, but every setup we consider is indeed in-context learning.
> >
> > > Why is working out a linear projection for a “new point” not a new task (and not ICL) where as working out the projection of a point in a new linear regression problem ICL?
> >
> > Great question. As noted above, regressing a new point but maintaining a single linear regression function implies that the corresponding function class $\mathcal{F}$ consists of a single member. In-context learning typically implies that the model is generalizing to unseen, novel functions in $\mathcal{F}$, which is impossible if $\mathcal{F}$ contains only a single member. In this way, classical linear regression may not be ICL, but learning the linear regression algorithm for arbitrary functions would certainly be.
> >
> >
> > >Did you consider trying to get even more "meta" so each X,Y pair represents a type of regression problem?
> >
> > This is a really interesting point, and we’re unaware of any literature that interrogates this level of “meta-learning” in an ICL setting. These are all great points we will summarize in our manuscript, and will certainly consider pursuing further. Thanks for a great discussion!

---

> > ### Comment · Reviewer_v7Gh · 2024-11-15
> > **Auto-Regressive Questions**
> >
> > Thanks for your quick reply.
> >
> > I now realise maybe I wasn't specific enough with some of my questions regarding the details of how the different classes of model were being used, and what I meant by auto-regressive. Hopefully the below questions are clearer.
> >
> > 1) For Figure 5 I assume the Auto-Regressive loss, detailed in the caption was used during training as well as evaluation?
> >
> > 2) For the results in the main paper what losses we used to train the different models? For the transformer and Mixer was the loss only applied to the final token X^{(-1)} or is the loss applied to all tokens, in a sequence to sequence manner? If only the final X was used, do you know if computing the other tokens factored into the FLOP count?
> >
> > 3) For Figure 5 how long was the context L for the first and last element in the sum?
> >
> > 4) Am I right in saying all models required 1 forward pass during training and evaluation, excluding figure 5 when all models required the same fixed number of forward passes during training and evaluation?
> >
> > 5) In section C.4 you mention the transformer uses a "causal attention mask", why did you use a causal attention mask? This is what I meant about the model being auto-regressive, why restrict the transformer attention in this way when the other models do not have a similar mechanism. Depending on your answers to question 1-4 I agree this might not make much difference in practice.
> >
> > 6) In any of the experiments were the models ever predicting x's? Were any of the loss functions ever applied to x's?
> >
> > I think adding some more details here would give readers greater clarity that the efficiency comparison between the different sorts of model was as fair as possible.
> >
> > Thank you for your other answers.

---

> > > ### Author Response · Authors · 2024-11-15
> > > **Auto-regressive clarifications**
> > >
> > > Thanks for the questions! We’ll definitely update our manuscript to clarify these points. Answering below:
> > >
> > > >For Figure 5 I assume the Auto-Regressive loss, detailed in the caption was used during training as well as evaluation?
> > >
> > > Yes, the auto-regressive loss was used in both training and testing.
> > >
> > > >For the results in the main paper what losses were used to train the different models? For the transformer and Mixer was the loss only applied to the final token X^{(-1)} or is the loss applied to all tokens, in a sequence to sequence manner? If only the final X was used, do you know if computing the other tokens factored into the FLOP count?
> > >
> > > For the non-autoregressive objectives, MSE is used in regression tasks and cross entropy in classification tasks. As you point out, the loss is computed only against the final token, and *not* for any intermediate tokens (unless using an auto-regressive objective). Intermediate tokens in the final layer do contribute to the FLOP count during a forward pass, but not during the backward pass since only the final token is used in the loss. One could architecturally modify the Transformer or Mixer to discard intermediate tokens in the final layer on non-autoregressive tasks, and thereby save a small amount of FLOPs in this setting. Since possible optimizations are endless, we’ve opted to avoid manual optimizations like this and used off-the-shelf vanilla implementations across all tasks.
> > >
> > > >For Figure 5 how long was the context L for the first and last element in the sum?
> > >
> > > We use the same max context length $L = 8$ as in the main text (except of course in Fig 5b, where $L$ varies). We start with a minimum context length of 2 points, since linear regression on fewer than 2 points is meaningless.
> > >
> > > >Am I right in saying all models required 1 forward pass during training and evaluation, excluding figure 5 when all models required the same fixed number of forward passes during training and evaluation?
> > >
> > > Yes, all models process a single forward pass on non-autoregressive tasks.
> > >
> > > >In section C.4 you mention the transformer uses a "causal attention mask", why did you use a causal attention mask? This is what I meant about the model being auto-regressive, why restrict the transformer attention in this way when the other models do not have a similar mechanism. Depending on your answers to question 1-4 I agree this might not make much difference in practice.
> > >
> > > Great questions. Our Transformer is indeed a decoder-only Transformer, maintaining continuity with [Raventos et al](https://arxiv.org/abs/2306.15063) and [Reddy](https://arxiv.org/abs/2312.03002), from which we’ve adapted the regression and classification tasks respectively. For our non-autoregressive tasks, as you suggest, this does not make much difference since the model sees the entire input before responding. For the auto-regressive task, this allows us to continue using the same architecture for these experiments.
> > >
> > > >In any of the experiments were the models ever predicting x's? Were any of the loss functions ever applied to x's?
> > >
> > > No, none of the loss functions were ever applied to the input points $x$. In the auto-regressive experiments, the model only ever predicts regression outputs $f(x_i)$ for varying context lengths, but never needs to predict the $x_i$ themselves.

---

> ### Comment · Reviewer_v7Gh · 2024-12-03
>
> Thank you for answering the computational comparison questions your choices do indeed sound the most reasonable given the clarifications. In light of your response I will raise my score from a 5 to a 6, and the soundness to a 4. While the work is of undoubtedly of high quality and thought provoking, I believe the overall contribution is still limited given the scope of the toy problems considered, and thus I feel unable to raise further.

---

### Official Review · Reviewer_2Mnt · 2024-11-04

**Soundness:** 3
**Presentation:** 4
**Contribution:** 3
**Rating:** 8
**Confidence:** 4

**Summary:**

This paper studies the in-context learning capabilities of MLPs and mixer-MLP models, comparing them to Transformers on tasks such as synthetic regression and classification. The models are evaluated on limited training data and larger test sets to determine when they shift from in-weight learning to in-context learning. The authors also introduce unique relational tasks—match-to-sample, sphere oddball, and line oddball—revealing that MLPs and relationally bottlenecked MLPs outperform Transformers on these tasks. They suggest that the results may stem from the inductive biases of these architectures.

**Strengths:**

1. Every experiment in the paper is designed thoroughly.
2. This is the first work encountered that explores the ICL capabilities of MLPs, which could be relevant to the literature on foundation models, especially in time series.
3. The addition of relational tasks to the existing synthetic regression and classification experiments contributes valuable insights into Transformer limitations. Transformers perform poorly when test exemplars differ significantly from the training data.

**Weaknesses:**

The paper could have included real regression data. Most existing literature focuses on synthetic tasks, and exploring real data (even simple regression datasets) with somewhat complex underlying distributions would have added valuable insights.

**Questions:**

Can you provide more insights on why transformers are failing in relational tasks?

---

> ### Author Response · Authors · 2024-11-15
>
> Thank you for the comments and suggestions! We’re glad you found the study to be thorough, impactful, and convincing. Answering your questions:
>
> > The paper could have included real regression data. Most existing literature focuses on synthetic tasks, and exploring real data (even simple regression datasets) with somewhat complex underlying distributions would have added valuable insights.
>
> Great suggestion. Our focus was primarily on measuring in-context learning in a controlled setting, leading us to synthetic ICL tasks. Exploring naturalistic regression data sounds like an excellent next step to examining more complex settings.
>
> > Can you provide more insights on why transformers are failing in relational tasks?
>
> Great question. To clarify this result, Transformers in fact succeed quite beautifully on relational tasks. However, comparatively smaller MLPs also succeed beautifully, and (surprisingly) generalize better on out-of-distribution perturbations. It remains unclear where this compute gap originates, but is a topic of great interest under study in a separate, theory-oriented follow-up we are preparing. We will update our manuscript to clarify this nuance.

---

> > ### Comment · Reviewer_2Mnt · 2024-11-27
> > **Final Comments**
> >
> > Thank you for addressing my concerns. While the paper is well-written and provides a fair comparison of MLPs and Transformers on their ICL capabilities in cleverly designed experiments, the paper's impact could have been stronger with evaluation on real datasets, as results in a controlled setting do not always generalize to real-world scenarios. But given the merits of the paper, I will retain my score.

---

### Official Review · Reviewer_qhqq · 2024-11-06

**Soundness:** 3
**Presentation:** 3
**Contribution:** 2
**Rating:** 8
**Confidence:** 4

**Summary:**

This paper provides more evidence that in-context learning (ICL) is a unique capability of Transformer models by demonstrating that Multi-Layer Perceptrons (MLPs) and MLP-Mixer models can also effectively learn in-context. The authors show that:

1) On well studied regression and classification ICL tasks , MLPs perform competitively with Transformers when given the same compute budget.
2) On classical relational reasoning tasks from psychology, MLPs actually outperform Transformers both in compute efficiency and out-of-distribution generalization. This challenges prior beliefs about MLPs' limitations in relational reasoning.
3) The authors demonstrate that MLPs, like Transformers, show a transition from in-weight learning to in-context learning as data diversity increases across their experimental tasks.

The findings support the growing interest in Transformer alternatives, and studies there capabilties in controlled synthetic tasks  while acknowledging that their findings align with existing results showing MLPs' competitiveness on more complex natural language and vision tasks.

**Strengths:**

The paper is quite well writtin and in my opinion easy to follow. The experiments seems well executed and believable, some questions remain, see below.

**Weaknesses:**

The paper in my opinion overclaims the signifiance of the work, of how surprising the findings are. MLPs are universal function approximators, and ofc, can to some extend approximate self-attention layers. Its nevertheless somewhat interesting that gradient descent can install such solutions into architectures purely consisting of MLPs.
It is, especially on tractable problems such as linear regression / classification, clear that, if optimized well, neural networks will find / approximate the (known) Byaes optimal solution of these problems. I therefore not find the results very surprising.

I would benefit the authors to highlight that Transformers architectures dynamically allocating compute / memory based on its in-context length. This is a unique feature, when comparing to RNNs or MLPs. Even if they might be performing similarlry to MLPs for a given sequence length given a fixed memory, compute budget, the flexiblity of Transformers is their strength.

The authors, afaiu, do not run experiments in an autoregressive model as e.g. Garg et al., 2022 (What Can Transformers Learn In-Context? A Case Study of Simple Function Classes) or von Oswald et al. 2023 (Uncovering mesa-optimization algorithms in Transformers). I find this setting very important, see Questions below.

**Questions:**

1) Can you please provide additional experiments and provide analyses of these results when training autoregressively as Garg et al., 2022 or  von Oswald et al. 2023.

2) If the MLPs / MLP mixers networks are chaning from in-weights to in-ciontext learning, do they approximate  e.g.  gradient descent in their archicture or mimic functionally of the self-attention layers. For the MLP mixer variants, I would find an analyses of functional similarity between architectures interesting. Can you e.g. train read-out layers at different depths of the trained network which approximate the solution / optimal prediction similarly when going down the network. It would be interesting to see if the networks are comptuing similar things at the same depth of the network (and gradually approximating the final solution). One could even try to see if the output of the self-attention and MLP mixers are similar.

---

> ### Author Response · Authors · 2024-11-15
>
> Thanks for the comments and suggestions! We’re glad you found our paper to be well-written and convincing. Answering your questions:
>
> > The paper in my opinion overclaims the signifiance of the work, of how surprising the findings are. MLPs are universal function approximators, and ofc, can to some extend approximate self-attention layers. Its nevertheless somewhat interesting that gradient descent can install such solutions into architectures purely consisting of MLPs. It is, especially on tractable problems such as linear regression / classification, clear that, if optimized well, neural networks will find / approximate the (known) Byaes optimal solution of these problems. I therefore not find the results very surprising.
>
> Thanks for raising an interesting point of discussion. MLPs are indeed universal function approximators, though it remains unknown at what cost – solving an in-context problem might require an intractably wide MLP. It is also unclear, as you note, whether an appropriate solution can be discovered by gradient descent. Given the field’s focus on attention-based architectures, it was very surprising to us that an MLP can indeed learn in-context with compute comparable to a Transformer’s, especially on tasks that are most heavily studied using attention-based constructions.
>
> > I would benefit the authors to highlight that Transformers architectures dynamically allocating compute / memory based on its in-context length. This is a unique feature, when comparing to RNNs or MLPs. Even if they might be performing similarlry to MLPs for a given sequence length given a fixed memory, compute budget, the flexiblity of Transformers is their strength.
>
> This is a great point, and we will add it to our manuscript. Thanks!
>
> > The authors, afaiu, do not run experiments in an autoregressive model as e.g. Garg et al., 2022 (What Can Transformers Learn In-Context? A Case Study of Simple Function Classes) or von Oswald et al. 2023 (Uncovering mesa-optimization algorithms in Transformers). I find this setting very important, see Questions below.
>
> Our apologies, we do indeed run experiments in an autoregressive setting (Fig. 5), and show that the exact same qualitative results hold. We will clarify this point further in our manuscript, and consider moving this figure out of the appendix.
>
> > Can you please provide additional experiments and provide analyses of these results when training autoregressively as Garg et al., 2022 or von Oswald et al. 2023.
>
> Gladly! The results of these experiments were discussed in Fig. 5, and show that the exact same results continue to hold in an autoregressive setting. We will clarify our manuscript to make these results easier to find.
>
> > If the MLPs / MLP mixers networks are chaning from in-weights to in-ciontext learning, do they approximate e.g. gradient descent in their archicture or mimic functionally of the self-attention layers. For the MLP mixer variants, I would find an analyses of functional similarity between architectures interesting. Can you e.g. train read-out layers at different depths of the trained network which approximate the solution / optimal prediction similarly when going down the network. It would be interesting to see if the networks are comptuing similar things at the same depth of the network (and gradually approximating the final solution). One could even try to see if the output of the self-attention and MLP mixers are similar.
>
> These are really interesting suggestions to examine, thank you! You raise a great question in asking mechanistically what an MLP may be doing when it performs a task in-context. Mechanistic interpretability falls outside the scope of this particular study, but we are currently working on a mechanistic follow-up and will consider incorporating your proposed methods. Thanks!

---

> > ### Comment · Reviewer_qhqq · 2024-11-29
> > **Thank you**
> >
> > Thank you for the clarifications. I will increase my score as is but vote for accepting the paper as I think the study is of high quality and is of interest to the ICLR audience. Many thanks!

---

### Official Review · Reviewer_KD93 · 2024-11-07

**Soundness:** 3
**Presentation:** 3
**Contribution:** 3
**Rating:** 6
**Confidence:** 4

**Summary:**

This paper presents experiments to argue that MLPs and MLP-Mixers are almost as effective as Transformer on many in-context learning (ICL) problems. Experiments in various settings like in context linear regression, classification and relational tasks from the field of psychology like sample match (akin to nearest neighbors in context), and finding the odd example out. In all of these settings, the paper presents experiments showing that MLP, MLP-Mixers and Transformers, by and large, perform similarly on these tasks at the same FLOPs spent on training. In particular, MLPs are slightly worse on the standard ICL problems, whereas they are better on the psychology inspired problems. Overall, through synthetic experiments, the paper makes that case that architectural biases may not play a huge role for ICL with enough compute.

**Strengths:**

1. This is the first paper, to my knowledge, that highlights that MLPs alone can lead to incontext learning. This is an interesting finding since, at least intuitively, the belief is that self attention helps with ICL. Verification of in-weight to in-context transition with task diversity, for all architectures, was also an interesting finding

2. Presentation and discussion of results is clear

3. For the set of ICL problems considered, the analysis seems quite extensive

**Weaknesses:**

1. The analysis is mostly in stylized and restricted settings. It is not entirely clear what this means for kinds of ICL that is observed in realistic settings (this is also mentioned in the limitations section of the paper). Even within simplistic settings, some more complex problems can be considered to make the claim that MLPs are competitive with Transformers. See questions 5 and 6 below.

2. Some useful description of the experimental setup, like input distribution,  how MLP and MLP-Mixer were used, were either missing or in the appendix. There is also some theoretical discussion in the appendix that are not referred to in the main paper. See question 1 below.

3. The paper mostly shows empirical evidence that MLPs can be competitive, however there is not much discussion about why this might be the case. See questions 2, 4 below


On the whole I still think the contributions are positive, but they could have been more extensive.

**Questions:**

1. What is the distribution of inputs $x_i$ for Section 2.1 What could happen if the tasks were even more diverse by changing the data covariance? This is important because for a fixed input covariance, even a 1-layer Transformer suffices to solve incontext linear regression.

2. Any understanding/analysis of why Transformers fair poorly in the relational tasks and why MLPs might be better? These seem right up the alley for Transformers, especially match-to-sample since attention directly computes all inner products.

3. Are there previous papers that argue that attention is required for ICL?

4. These experiments mix the role of expressivity (how large the model needs to be to solve this task) and optimization (can standard algorithms learn a good solution fast enough).

5. Any reason to look at linear regression/classification and not other ICL (like decision trees, or fitting MLPs like considered in Garg et al.)? It raises the question whether the findings are an artifact of simplicity of the chosen problems.

6: One strength of Transformers is the ability to handle different context lengths n with a single model. Is this property also true for MLPs?

---

> ### Author Response · Authors · 2024-11-15
>
> Thanks for the comments! We’re glad you found our results compelling.
>
> >The analysis is mostly in stylized and restricted settings. It is not entirely clear what this means for kinds of ICL that is observed in realistic settings. Even within simplistic settings, some more complex problems can be considered
>
> Great point. We were interested in cleanly evaluating ICL performance, and sought to avoid confounds introduced in naturalistic settings like developing fluency in natural language or learning complex visual features (which are certainly important topics of study in their own right, but outside the scope of our investigation). Hence, we adopted synthetic tasks common in the literature. The tasks you reference are indeed quite interesting, and there are an ever-growing collection of fantastic tasks to try. In our case, we found that regression, classification, match-to-sample, sphere oddball, line oddball, and same-different were a sufficiently diverse and complex collection to make our point. In subsequent follow-ups, we will be delighted to examine other settings further.
>
> > Some useful description of the experimental setup were missing or in the appendix. There is some theoretical discussion in the appendix that are not referred to in the main paper.
>
> Great catch. We apologize for the gaps, and will correct them in our manuscript.
>
> > The paper mostly shows empirical evidence that MLPs can be competitive, however there is not much discussion about why this might be
>
> Great point. We comment further below.
>
> > What is the distribution of inputs $x_i$ for Section 2.1? What could happen if the tasks were even more diverse by changing the data covariance?
>
> Inputs $x$ are sampled iid from a standard Gaussian. Changing the data covariance sounds like a fascinating direction for future study, thanks for the suggestion!
>
> > Any understanding of why Transformers fair poorly in the relational tasks and why MLPs might be better? These seem right up the alley for Transformers
>
> Great question. To clarify, the Transformer does succeed beautifully at match-to-sample, as well as the relational tasks generally. However, comparatively smaller MLPs succeed beautifully as well, and (surprisingly) generalize better out-of-distribution. This study was intended to be primarily empirical, documenting surprising performance gaps. We are currently preparing a separate, related follow-up with theoretical arguments.
>
> > Are there previous papers that argue attention is required for ICL?
>
> There are many. [Akyurek et al](https://arxiv.org/abs/2211.15661), [Von Oswald et al](https://arxiv.org/abs/2212.07677) and [Zhang et al](https://arxiv.org/abs/2306.09927) are a few. As Von Oswald and colleagues put it, “Transformers show remarkable in-context learning behavior. Mechanisms based on attention, associative memory and copying by induction heads are currently the leading explanations for this remarkable feature of learning.” Many theories about ICL explicitly ground their constructions in attention-based architectures. One of our core motives is to encourage the field to move past this restriction, and consider ICL as a broader phenomenon.
>
> > These experiments mix the role of expressivity (how large the model needs to be) and optimization (can standard algorithms learn a good solution)
>
> That’s correct, and this is a deliberate design choice. We measure models based on floating point operations, sweeping over a wide range of sizes and training time. By budgeting based on FLOPs, a model cannot have both unbounded parameter count and arbitrarily large training time, forcing an optimal tradeoff. Doing so allows us to compare diverse architectures fairly and reflects the real-world cost of training these models.
>
> > Any reason to look at linear regression/classification and not other ICL tasks (like in Garg et al.)?
>
> Great question. The additional tasks in Garg et al. are certainly interesting, as are the ever-growing collection in the literature, and regrettably we must select a finite subset to try. In our case, we found that regression, classification, match-to-sample, sphere oddball, line oddball, and same-different were sufficiently diverse and complex to make our point. Additional perturbations like varying data diversity, increasing context length (Fig. 1) or the various out-of-distribution settings (Fig. 2) probe whether our models have learned a shortcut that leverages the simplicity of the tasks, or are truly performing in-context. We have found the latter to be true, and look forward to confirming this finding with additional tasks in the future.
>
> > One strength of Transformers is the ability to handle different context lengths n. Is this property also true for MLPs?
>
> Yes, this is also true for MLPs. We consider variable-length inputs in Fig. 5, where the models are prompted auto-regressively. For the MLPs, inputs are zero-padded up to a maximum context length. We show that the exact same results continue to hold in this setting.

---

### Meta-Review · Area_Chair_hrye · 2024-12-24

**Metareview:**

This paper studies the in-context learning abilities of MLP. Such property has been recently been studied extensively for Transformer models, however, it has not been explored for MLPs. Arguments for in-context learning of Transformers usually rely on the attention mechanism of Transformers. So it is indeed interesting that they exhibit such abilities.

All the reviewers note that the paper is interesting and well presented. One of the reviewers was concerned about ICL ability of MLP not being competitive to Transformers in real-world settings. This is a valid concern. As noted in the paper, ICL ability of MLPs are not necessarily powerful in all settings but nevertheless I think it is an valuable to have the wider community know about these results to enable better understanding of this area. While the authors discuss the limitations of the work, I would like them to further highlight the limitations of this work especially in the real-world settings (both in abstract & conclusions). I recommend acceptance conditioned on this.

**Additional Comments On Reviewer Discussion:**

The authors have done a commendable job in addressing the key concerns of the reviewers. After the rebuttal, most of the reviewers were positive about the paper. One of the reviewers was concerned about the claims of the paper. As discuss in the metareview, while this concern is valid, I think this can be addressed fairly easily in the final version.

---

### Decision · Program_Chairs · 2025-01-22

Accept (Poster)